# High-fidelity topochemical polymerization in single crystals, polycrystals, and solution aggregates

Chongqing Yang [1,9], Jianfang Liu [1,9], Rebecca Shu Hui Khoo [1], Maged Abdelsamie[1,8], Miao Qi [1], He Li [1,2], Haiyan Mao[3], Sydney Hemenway [1], Qiang Xu [4], Yunfei Wang[1,5,6], Beihang Yu [1], Qingsong Zhang[1], Xinxin Liu[7], Liana M. Klivansky[1], Xiaodan Gu [6], Chenhui Zhu[5], Jeffrey A. Reimer [3], Ganglong Cui [7], Carolin M. Sutter-Fella [1], Jian Zhang [1], Gang Ren [1] & Yi Liu [1,2] ✉

Topochemical polymerization (TCP) emerges as a leading approach for synthesizing single crystalline polymers, but is traditionally restricted to transformations in solid-medium. The complexity in achieving single-crystal-to-single-crystal (SCSC) transformations due to lattice disparities and the untapped potential of performing TCP in a liquid medium with solid-state structural fidelity present unsolved challenges. Herein, by using X-rays as the primary means to overcome crystal disintegration, we reveal the details of SCSC transformation during the TCP of chiral azaquinodimethane (AQM) monomers through in situ crystallographic analysis while spotlighting a rare metastable crystalline phase. Complementary in situ investigations of powders and thin films provide critical insights into the side-chain dependent polymerization kinetics of solid-state reactions. Furthermore, we enable TCP of AQM monomers in a liquid medium via an antisolvent-reinforced aggregated state, yielding polymer nanofibers with high crystallinity akin to that of solid-state. This study testifies high structural precision of TCP performed in different states and media, offering critical insights into the synthesis of processable nanostructured polymers with desired structural integrity.

The synthesis of single crystalline polymers stands as a crucial yet intricate endeavor, pivotal for unraveling the fundamental relationship between structure and chemical/physical properties at molecular level[1–4]. Topochemical polymerization (TCP) emerges as an encouraging protocol enabling the production of macroscopically ordered polymers, via a lattice-controlled approach under external stimuli[5–8]. The prospect of single-crystal-to-single crystal (SCSC) transformation distinguishes TCP as a rare avenue for producing single crystalline polymers, yet

[1]The Molecular Foundry, Lawrence Berkeley National Laboratory, Berkeley, CA 94720, USA. [2]Materials Sciences Division, Lawrence Berkeley National Laboratory, Berkeley, CA 94720, USA. [3]Department of Chemical and Biomolecular Engineering, University of California, Berkeley, CA 94720, USA. [4]Chemical Science Division, Lawrence Berkeley National Laboratory, Berkeley, CA 94720, USA. [5]Advanced Light Source, Lawrence Berkeley National Laboratory, Berkeley, CA 94720, USA. [6]School of Polymer Science and Engineering Center for Optoelectronic Materials and Devices, The University of Southern Mississippi, Hattiesburg, MS 39406, USA. [7]Key Laboratory of Theoretical and Computational Photochemistry, Ministry of Education, Chemistry College, Beijing Normal University, Beijing 100875, P.R. China. [8]Present address: Interdisciplinary Research Center for Intelligent Manufacturing and Robotics, King Fahd University of Petroleum and Minerals (KFUPM), Dhahran 31261, Saudi Arabia. [9]These authors contributed equally: Chongqing Yang, Jianfang Liu. ✉e-mail: yliu@lbl.gov

achieving a SCSC transformation during solid-state polymerizations remains nontrivial[9-17]. In general, the initiation of the SCSC process requires aligning the reactive groups of monomer crystals in parallel within a maximal separation of 4.2 Å (adhering to the Schmidt criterion)[18]. Nevertheless, intriguing occurrences arise wherein monomer crystals display no TCP reactivity despite their perfect alignment fulfilling the geometrical requirement[19,20], while others, though against these guidelines, exhibit notable reactivity[21,22]. Recent studies suggest that external stimuli can induce/augment molecular motion within lattices[23-25], changing the distances between monomer reactive groups to a topotactic geometry that impacts their original TCP reactivity[26]. Therefore, investigating the monomer motion and subsequent polymerization process under stimuli becomes pivotal in unraveling the authentic SCSC transformation mechanism. However, direct observation of the SCSC evolution process remains rare, due to the lack of characterization techniques suitable for capturing this transient and dynamic transformation[27]. Additionally, significant strain can accumulate within lattice during the TCP process, easily causing the disintegration of crystals, especially in systems involving large lattice deformation or fast reaction kinetics[5,28,29]. Therefore, proficient strain control becomes a critical but arduous necessity to achieve polymers with a large single crystalline domain via SCSC transformation. The successful implementation of this will yield deeper insights into the complex dynamics of the TCP process.

As TCP reactions are typically restricted to solid-state processes, they face inherent limitations such as poor processability. Replicating the additive-free, and facile TCP reaction in less confined medium such as liquids, would open up a new paradigm for engineering processable, crystalline polymeric materials. However, in such environments where lattice reinforcement is usually absent, the feasibility and integrity of TCP are markedly compromised. To date, the sparse examples of TCP in liquid-medium predominantly rely on functionalized diacetylene monomers equipped with necessary auxiliary directing groups (i.e., multiple hydrogen bonds) to aid monomer preorganization[30-37]. It is therefore imperative to devise effective strategies for promoting the assembly of other TCP-capable molecules in a liquid medium to enable topochemical polymerization. Subsequent structural validation, ideally approaching atomic resolution, is also critical to corroborate the fidelity of polymerization processes in the absence of lattice confinement as seen in crystals.

Herein we devised a series of chiral quinoidal monomers for in-depth elucidation of the TCP process across various media, including single crystals, polycrystals, thin films, and aggregates in solution. By employing X-ray as the sole irradiation source, crystal disintegration was minimized despite significant lattice deformations, thereby achieving SCSC transformation. In situ crystallographic studies unveiled the structural evolution involving a rare metastable phase within the precursor crystals, which played a crucial role in preserving the integrity of the resulting polymer crystal by partially mitigating mechanically induced strain. The findings offer unexplored details on the structural changes occurring during TCP, aligning with insights obtained from in situ studies of TCP in thin films and polycrystalline powders. Moreover, we successfully demonstrated the aggregation-reinforced assembly of precursor molecules in liquids, paving the way for TCP reactivity akin to that observed in the solid state, leading to the formation of colloidal single crystalline polymer nanofibers. The individual polymer chain structures of these highly crystalline nanofibers were elucidated using cryogenic electron microscopy (cryo-EM), thereby substantiating the feasibility of achieving high-fidelity TCP in a liquid medium.

## Results and discussion

### Design and arrangement of chiral monomers within molecular single crystals

Our study is centered around azaquinodimethanes (AQMs)[38], a recently recognized class of compounds with thermally and photochemically

induced linear chain polymerization activity[39]. These molecules exhibit a propensity to pack in close stereochemical proximity in the solid state, facilitating polymerization. Despite the established TCP reactivity of AQMs, the attainment of polymer single crystals with adequate size and quality for single-crystal X-ray diffraction (SCXRD) analysis remains challenging. In an effort to unravel the SCSC evolution in both dense (solid state) and less confined media (liquid), AQM monomers with chiral side chains were devised to tune the specific molecular arrangement, assembly and chiroptical properties. As shown in Fig. 1a, the AQM molecules CM1-s/r and CM2-s/r bearing one and two chiral 2-methylbutoxy side groups on each phenyl end groups, respectively, were obtained as enantiomerically pure pairs of isomers (Supplementary information). All monomers readily crystallized and formed needle-like single crystals from chloroform. SCXRD analysis indicated that the conjugated core of all the AQM molecules adopted a nearly coplanar conformation, which arranged into slipped columnar stacks with a π-π distance of ~3.34 Å. Both CM1-s/r enantiomers packed in a tetragonal chiral $P2_1$ space group with two crystallographically independent molecules per unit cell (Fig. 1b, c, Supplementary Fig. S1 and Supplementary Tables S1-2). Notably, the 2-methylbutoxy side chains of adjacent molecules within the same column bore different conformations, in which the methyl groups were oriented differently. This inequivalence results in an AB pattern along the column axis, commensurate with two non-identical $d_{cc}s$ (distances between the reactive methylene carbons on adjacent monomers) of 3.67 and 3.63 Å. In contrast, the CM2-s/r enantiomers packed within a chiral P1 group, as depicted in Fig. 1c, d (see also Supplementary Fig. S2). They displayed a higher-symmetry AA stacking pattern within individual π-stacked columns, yielding a uniform $d_{cc}$ of ~3.65 Å.

All monomer crystals readily polymerized when exposed to visible light irradiation or heat, which underwent characteristic color changes from red (CM1-s/r) /yellow (CM2-s/r) to nearly colorless, affording polymers CP1-s/r and CP2-s/r (Fig. 1d-f and Supplementary Fig. S3). Given the nearly identical packing geometries observed in each pair of chiral enantiomers, CM1-s and CM2-s were chosen as representatives to illustrate their SCSC TCP behaviors. Fourier transform infrared (FTIR) spectra of CP1-s and CP2-s (Supplementary Figs. S4-5) revealed the disappearance of quinoidal C = C stretching peaks centered at 1597 cm$^{-1}$ and 1625 cm$^{-1}$, respectively[40], indicating the successful completion of the TCP reaction. As further confirmed by cross-polarization/magic angle spinning (CP/MAS) solid-state $^{13}$C NMR ($^{13}$C-SSNMR) spectroscopy (Supplementary Figs. S6-7), the emergence of distinct methine peaks centered at ~50 ppm was indicative of the newly formed C-C bonds from connecting the adjacent methylene carbons.

### Structural evolution during SCSC polymerization

When single crystals of CM1-s and CM2-s were subjected to visible light irradiation during SCXRD data acquisition, even at temperatures as low as 100–200 K, polymerization proceeded but with significant crystal deformation and the loss of single crystal quality (Supplementary Fig. S8 and Supplementary Movie S1). We note that visible light irradiation only led to low quality crystals. Compared to commonly used light irradiation, X-ray (and gamma-ray) ionization radiation sources have a higher penetration depth that facilitates more homogeneous reactions throughout the crystal. More importantly, high-energy photons likely initiate polymerization through secondary processes derived from inelastic scattering or core electron excitations, rather than direct valence electron excitation as seen in UV-visible light irradiation. This makes X-ray irradiation more effective in facilitating polymerization while mitigating crystal disintegration[41,42]. At 200 K, it was found that continuous X-ray irradiation successfully induced SCSC polymerization (Fig. 2a–c). Remarkably, a metastable intermediate crystal phase (denoted as CM1*-s) was observed during ~3 h of X-ray irradiation at 200 K, which was followed by gradual

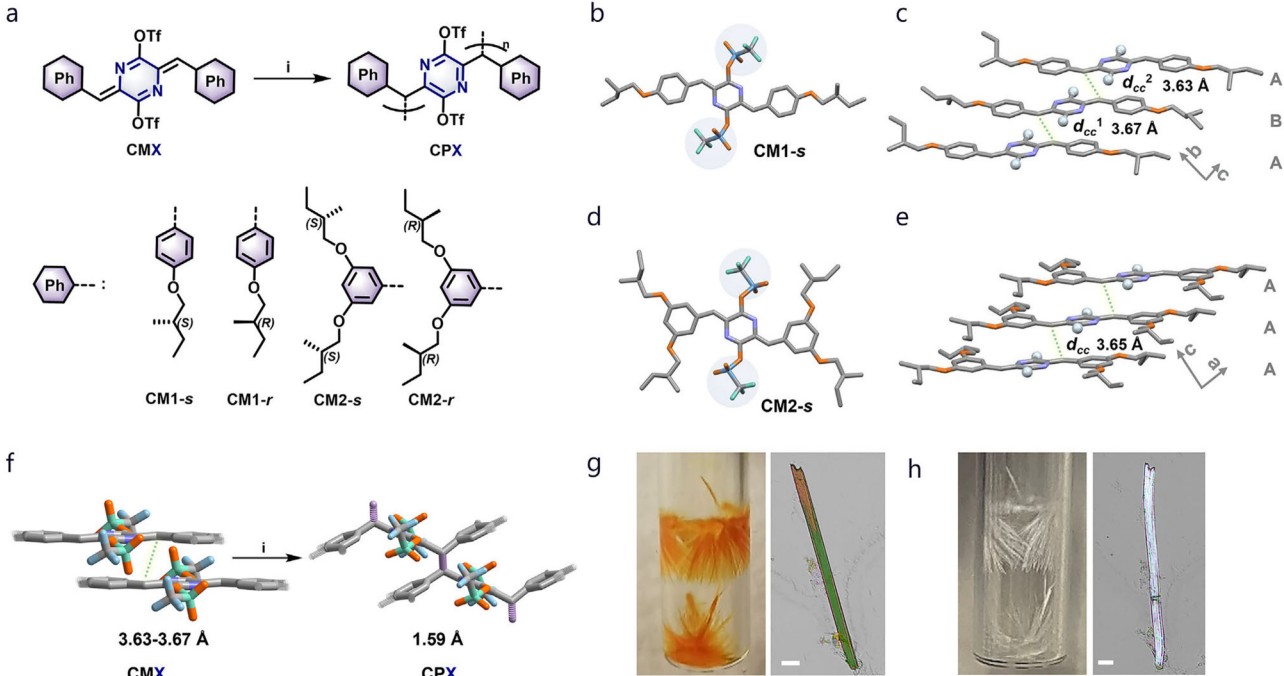

**Fig. 1 | Single crystal polymerization of single-chirality AQM monomers.**
**a** Synthetic route from chiral AQM monomers CMX to corresponding topochemical polymers CPX (X: 1-*s*, 1-*r*, 2-*s*, 2-*r*). Reaction condition *i*. light, heat or X-rays. **b** Single crystal X-ray structure of CM1-s. Atom color scheme: C (grey), N (light purple), O (orange), S (blue), F (green). H atoms are omitted for clarification. **c** Columnar packing of CM1-*s* with an AB pattern in crystals. Cyan balls represent the triflate groups for clarity. d$_{cc}$: the distance between reactive methylene carbon of neighboring molecules. **d** Single crystal X-ray structure of CM2-*s*. Atom color scheme: C (grey), N (light purple), O (orange), S (blue), F (green). H atoms are omitted for clarification. **e** Columnar stacking of CM2-*s* with an AA pattern in crystals. **f** Schematic illustration of topochemical polymerization of AQM cores under different triggers *i*. The C-C bond distance was reduced to ~1.5 Å after topochemical polymerization. **g** Photograph and optical microscopy of CM1-*s* single crystals grown from chloroform. Scale bar: 40 μm. **h** Photograph and optical microscopy of CP1-*s* single crystals from the polymerization of CM1-s single crystals shown in *g*. Scale bar: 40 μm.

transformation to the final polymer crystal phase within 4 h (Fig. 2d, e), as elucidated by in situ X-ray structure analysis.

In comparison with CM1-*s*, the crystal structure of CM1\*-*s* shares the same *P2₁* space group but contains only one molecule per unit cell (Supplementary Fig. S9). Accordingly, the CM1\*-*s* crystal has nearly half the cell volume as CM1-*s* (3212.9 vs 1608.6 Å³) (Supplementary Tables S3-4). The observed crystallographic phase change corresponds to the switching of the packing pattern from AB in CM1-*s* to AA in CM1\*-*s*, resulting from the synchronized rotation of the methyl end groups around the chiral carbon center of the side chains of CM1-*s* in one of the AB layers (Supplementary Fig. S10). Such conformational changes induced by X-ray activation concomitantly lead to a more precisely packed geometry and equalized d$_{cc}$s of 3.65 Å (Fig. 2d). Density functional theory (DFT) calculations based on the crystal structures confirm that CM1-*s* is lower in energy than CM1\*-*s* by ~1.4 kcal mol⁻¹ (Supplementary Fig. S11). This thermodynamic preference is consistent with the experimental results where CM1-*s* is the only starting crystal phase and CM1\*-*s* exists only as an intermediate, which is further corroborated by differential scanning calorimetry (DSC) results (see discussions later).

Upon prolonged irradiation, the crystal structure evolves in accordance with the diastereo-specific TCP polymerization to give the single diastereomeric CP1-*s*, with a preserved *P2₁* space group and a deformed unit cell dimension (Fig. 2e and Supplementary Table S4). Following the formation of new C-C single bonds, the C-C distance decreases from 3.65 Å in the monomer to 1.59 Å in the polymer, resulting in notable lattice changes from CM1\*-*s*, amounting to a 3.8% increase in the total cell volume (Supplementary Table S4). Such lattice changes are considerably smaller than that of transforming CM1-*s* directly to CP1-*s*. This observation highlights the crucial role of the metastable phase in preconditioning molecular packing, which

probably attenuates the mechanical stress induced by lattice changes, thereby preserving the structural integrity of the resulting polymer crystals. In comparison, prolonged X-ray irradiation of CM2-*s* single crystals under similar conditions succeeded in inducing gradual TCP. However, no metastable intermediate phase could be observed during the polymerization, nor did the polymer crystal diffract well enough for structural resolution due to the loss of crystal integrity.

## TCP progression in polycrystals
The X-ray induced polymerization of the AQM monomers in polycrystalline forms were also verified using powder XRD (PXRD; Fig. 3a, b). During the initial scan (2.0 h) of CM1-*s* polycrystals under continuous X-ray irradiation, a new peak appeared at 6.6° (d = 1.34 nm), corresponding to the (020) facet of CP1-*s*. Subsequent scans revealed an additional peak at 5.9°, alongside a decrease in monomer diffraction peaks, indicating the formation of a transitional phase during TCP. In the absence of X-ray irradiation, the PXRD pattern remained unchanged, further confirming that X-ray irradiation induces the observed TCP (Supplementary Fig. S12). Notably, due to similarities in peak positions and diffraction patterns between CM1-*s* and CM1\*-*s*, it was insufficient to identify the intermediate CM1\*-*s* (Supplementary Fig. S13). However, PXRD patterns clearly showed the coexistence of both the monomer and polymer phases, verifying a heterogeneous polymerization process where monomer and polymer crystals grew in separate domains[43], which contradicts to the common notion that homogeneous polymerization is required for a successful SCSC transformation[29,44]. This finding was consistent with the in-situ video results (Supplementary Fig. S14), which revealed that polymerization was initiated in localized regions—primarily at the edges and defects of the single crystal—causing these regions to turn colorless within ~2 min of light exposure. These localized polymerization zones expanded

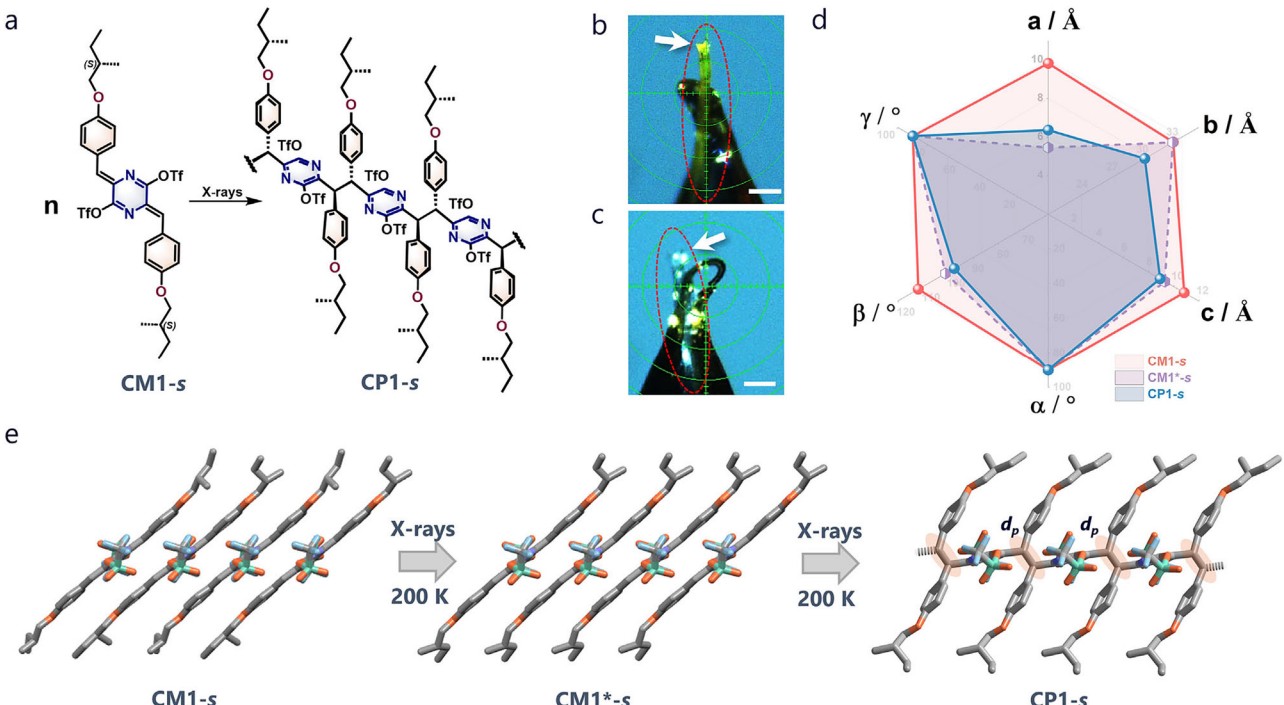

**Fig. 2 | X-ray induced SCSC polymerization of CM1-*s*. a** Chemical structures of CM1-*s* and CP1-*s*. Photographs of a CM1-*s* crystal mounted in diffractometer (**b**) before and (**c**) after X-ray induced polymerization (in the red circles). Scale bar: 1 mm. **d** The unit cell parameters of different phases during the SCSC TCP process of CM1-*s*. **e** Single crystal X-ray structures of CM1-*s*, CM1*-*s* and CP1-*s*.

over time, accompanied by visible crystal splitting. This observation strongly supported a heterogeneous mechanism for the AQM system. Similarly, in CM2-*s*, both monomer (diffraction at 6.0°; d = 1.47 nm) and polymer (diffraction at 6.5°; d = 1.36 nm) phases were clearly observed, suggesting a similar heterogeneous polymerization with phase separation.

The X-ray induced phase changes of CM1-*s* and CM2-*s* also correlate well with the thermally induced changes revealed by DSC. The exothermic peak of CM1-*s* at 89 °C indicated successful thermally-induced polymerization, with a ΔH of -7.8 kcal/mol (Fig. 3c). Notably, prior to polymerization, an endothermic peak at 62 °C was observed, signifying a crystal-to-crystal phase transition (ΔH ≈ 1.3 kcal mol$^{-1}$). This phase change is reversible, as indicated by an exothermic peak at 57 °C when the sample was allowed to cool down before reaching the polymerization temperature (Supplementary Figs. S15-16). The associated thermal energy change agrees well with the calculated difference between CM1-*s* and CM1-*s** (1.4 kcal mol$^{-1}$). The additional sharp endothermic peaks observed at -180 °C corresponded to a reversible crystal-to-crystal phase change of the polymer, as was corroborated by variable temperature PXRD studies of CP1-*s* (Supplementary Figs. S17-18). In the case of CM2-*s*, no endothermic peak prior to the polymerization event at 71 °C (ΔH of -8.0 kcal mol$^{-1}$) was observed, suggesting the absence of pre-polymerization phase changes (Fig. 3d and Supplementary Fig. S19). In-situ grazing incidence wide-angle X-ray scattering (GIWAXS) measurements further elucidated detailed structural changes. Heating a spun-cast thin film of CM1-*s* to 80 °C resulted in a sudden shift of the monomer's (020) facet from q = 0.40 to 0.43 Å$^{-1}$ (corresponding to d = 1.57 to 1.46 nm) after 7.4 s, followed by lattice contraction and expansion before polymerization (Fig. 3e,f). Notably, the displacement -0.43 Å$^{-1}$ correlated well with appearance of an additional peak in PXRD at -6.0° (d = 1.47 nm), indicative of substantial strain built-up in the monomer crystal prior to the polymer phase separation, consistent with the in situ SCXRD studies. Corresponding polymer phase started to appear after 24.0 s and reached completion after 31.4 s, with the polymer's (020) facet emerging at

0.48 Å$^{-1}$ (d = -1.31 nm). In CM2-*s*, the polymerization completed within 7.3 s following similar lattice expansion, suggesting a faster polymerization kinetic (Fig. 3g-h).

Aside from XRD-derived structural information which is area-averaged over a bulk sample, cryo-EM also provided single-chain structural details via direct imaging. High-resolution cryo-EM image of individual CP1-*s* fibers, and corresponding fast Fourier Transform (FFT) (Fig. 3i-k) revealed clearly extended polymer chains along the fiber axis. The polymer chains pack tightly in the orthogonal direction with an interchain separation of ~1.35 nm. This periodicity, shown as the (020) facet in the FFT (Fig. 3k) electron diffraction pattern, also corroborated well with the primary PXRD peak of CP1-*s* at 6.6°. Moreover, an additional lattice spacing of 1.32 nm was observed in the orthogonal direction in the FFT diffraction pattern, corresponding to the periodicity along the polymer main chain. The HRTEM image aligns well with the single crystal structure of CP1-*s* projected on the [001] axis, despite that the main-chain periodicity is twice of that determined by SCXRD. In addition, different inter-chain separations of 1.02, 0.83 and 0.57 nm were also observed (Supplementary Fig. S20), all maintaining the same main-chain periodicity of 1.32 nm, which was attributed to the projection of the same polymer crystals with different orientations.

**Quantitative assessment of TCP reactivity**

To gain more quantitative insight into the reactivity of AQMs in the solid state, in-depth kinetic studies were carried out employing spun cast thin films of AQM monomers (thickness: -50 nm; Supplementary Fig. S21) in a customized in situ UV-vis absorption setup. As shown in Fig. 4a–d and Supplementary Figs. S22-23, the characteristic absorption peak of the AQM chromophore in the range of 400–560 nm disappeared completely upon continuous irradiation due to polymerization, providing great contrast for kinetic studies. Results showed that polymerization of both CM1-*s* and CM2-*s* followed pseudo-first order kinetics at different temperatures (Supplementary Figs. S24–26). At 293 K, polymerization of CM2-*s* was completed within

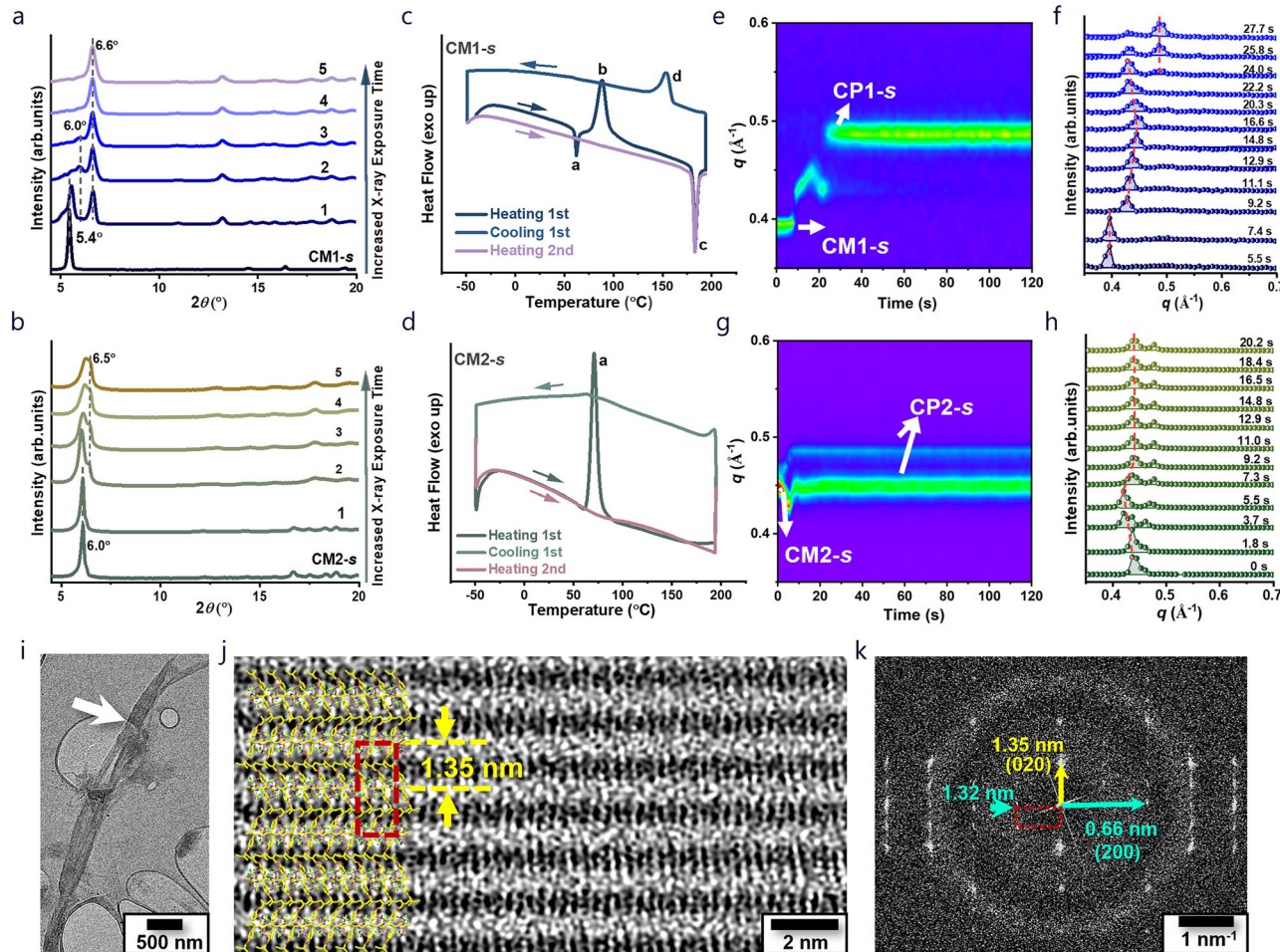

**Fig. 3 | Following TCP behavior in polycrystalline CM1-s and CM2-s.** Time-dependent powder XRD spectra of (**a**) CM1-*s* and (**b**) CM2-*s* under continuous X-ray exposure (λ = 1.54 Å). The numbers 1, 2, 3, 4 and 5 correspond to cumulative X-ray irradiation durations of 2, 4, 6, 8 and 10 hours, respectively. DSC studies of (**c**) CM1-*s* and (**d**) CM2-*s* at a heating rate of 10 °C/min. **e** False-color 2D GIWAXS plot of CM1-*s* during heating to 80 °C. **f** Corresponding 1D XRD patterns of CM1-*s*. **g** False-color

2D GIWAXS plot of CM2-*s* during heating to 80 °C. **h** Corresponding 1D XRD patterns of CM2-*s*. **i**, Cryo-EM image of polycrystalline CP1-*s* at low magnification. **j, k** High-resolution cryo-EM image and corresponding FFT of polycrystalline CP1-*s* at [001] axis orientation, showing the inter-chain separation of 1.35 nm and main-chain periodicity of 1.32 nm. The single-crystal model was superimposed on the image with a unit cell indicated by red dash line.

30 s, while for CM1-*s*, only about half reacted even after 350 s (Fig. 4c). The corresponding rate constant ($k$) for CM1-*s* was calculated to be $1.1 \times 10^{-3} s^{-1}$, ~52 times slower than that of CM2-*s* ($5.7 \times 10^{-2} s^{-1}$). Raising the temperature greatly accelerated the polymerization rate. At 333 K, CM1-*s* and CM2-*s* reached complete polymerization within 125 and 15 s with rate constants of $5.7 \times 10^{-2}$ and $1.9 \times 10^{-1} s^{-1}$, respectively. Notably, no appreciable monomer-polymer conversion was observed without light irradiation (Supplementary Figs. S27–29). These results confirm the negligible contribution of thermally induced polymerization at temperatures below 333 K, consistent with the DSC results. Correspondingly, an Arrhenius activation energy of 14.6 kcal mol⁻¹ was obtained for CM1-*s* from fitting the variable temperature kinetic data, in contrast to a much lower activation energy of 5.9 kcal mol⁻¹ for CM2-*s* (Supplementary Fig. S30). The disparity in reaction kinetics for the two monomers shows no correlation with their $d_{cc}$s, since CM2-*s* reacts ~50 times faster than CM1-*s* despite a very similar $d_{cc}$ of around 3.65 Å, verifying that close contact of reactive carbon atoms is essential for TCP but is irrelevant to the reaction kinetics[45]. Instead, the polymerization rate may be related to the difference in unit cell lattices between monomer and polymer, with larger changes leading to slower reaction rates. The additional energy required for the monomer-monomer phase transition of CM1-*s* may have also contributed to the reaction rate differences.

## Aggregation-reinforced TCP in liquid

Extending TCP reactivity from the solid state to a liquid medium represents an enticing yet challenging endeavor, given the lack of lattice reinforcement effects in conventional TCP monomers without directing groups. The combination of chirality and photo-reactivity in AQM molecules offers a unique avenue to probe the preorganization of AQM monomers and their TCP reactivity in liquid media (Fig. 5a). Fully dissolving the monomers in THF generated circular dichroism (CD) spectra with no Cotton effect in the AQM chromophore absorption range (250–550 nm), indicating insufficient preorganization of AQM molecules in solution. In contrast, the introduction of water as an antisolvent promoted the formation of supramolecular aggregates, enhancing AQM core interactions and resulting in observable Cotton effects in the CD spectra, reaching maximum intensity at 75% water content (Supplementary Figs. S31-32). Notably, mirror CD response was observed for the enantiomeric CM1-*r* and CM2-*r* under the same condition (Fig. 5b, c). These results confirm the aggregation-aided transfer of the molecular chirality to supramolecular assemblies, wherein the conjugated AQM cores of CM1-*s/r* and CM2-*s/r* molecules are well organized within an asymmetric environment imposed by the chiral side chains.

Facile polymerization was realized when the water/THF mixtures of CM1-*s* and CM2-*s* were subjected to photoirradiation, which resulted

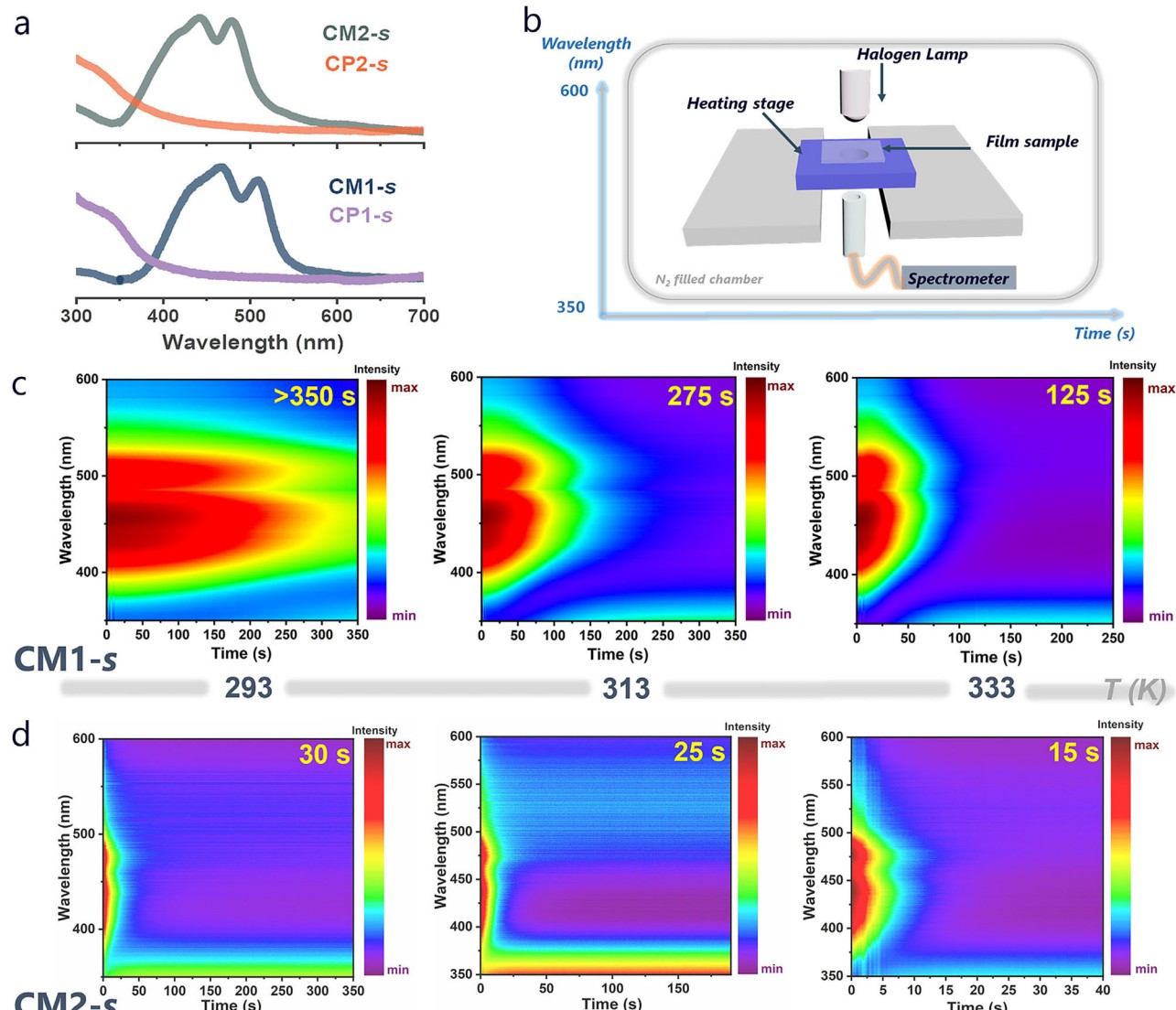

**Fig. 4 | Kinetic quantifications of AQM polymerization via in situ UV-vis study.**
**a** UV-vis spectra of thin films of CM1-*s* and CM2-*s*, and the corresponding CP1-*s* and CP2-*s* thin films after polymerization. **b** Schematic illustration of the customized in situ UV-vis absorption setup equipped with a temperature-controlling unit. False-color 2D plots of UV-vis scanning kinetic curves for (**c**) CM1-*s* and (**d**) CM2-*s* thin films at different temperatures. The numbers in the top right corner of the panels in **c** and **d** indicate the time needed to reach full polymerization.

in clear, colorless colloidal solutions showing Tyndall effect when exposed to light scattering test (Supplementary Figs. S33-34). Notably, TCP occurs within the aggregates dispersed in the transparent solution rather than in the fully dissolved monomeric state. CD spectra indicated that the strong bisignate peaks associated with the AQM chromophore disappeared, while weaker peaks appeared at 200–250 nm that corresponded to the chromophore in the nonconjugated polymer chain (Fig. 5b, c). Similar induced CD responses and polymerization behavior were observed for CM1-*r* and the CM2-*r* enantiomers. Dynamic light scattering (DLS) studies further revealed that the average hydrodynamic diameters ($D_h$) of CM1-*s* and CM2-*s* were determined to be ~300 nm and 199 nm, respectively (Supplementary Fig. S35). Following light irradiation, the $D_h$ values of CP1-*s* and CP2-*s* increased to 409 nm and 221 nm, respectively.

Cryo-EM and atomic force microscopy (AFM) analyses were then employed to provide direct morphological and structural information of the polymers. As shown in Supplementary Figs. 36-37, both polymers of CP1-*s* and CP1-*r* demonstrated nano-fabric morphologies with smooth surfaces. The widths of both nanofibers were in the range of 170-220 nm, and the minimum thickness were found to be ~28 nm and

~36 nm for CP1-*s* and CP1-*r* respectively. High-resolution cryo-EM images and FFT diffraction patterns of CP1-*s* revealed a main-chain periodicity of 1.30 nm along the fibril axis, as well as inter-chain separations of 1.35 nm and 1.03 nm, consistent with those observed in the fibers obtained from solid state polymerization (Supplementary Figs. S38-39). Based on the parameters obtained from cryo-EM, the CP1-*s* polymer chain packed in a lattice with P2₁ symmetry along the a-axis, with linear chain conformation similar to that determined by SCXRD analysis. In the case of CP2-*s* and CP2-*r*, smaller features were formed with widths in the ranges of 40-70 nm. Both fibers of CP2-*s* and CP2-*r* exhibited smooth surfaces with a thickness of ~16-17 nm (Fig. 5d and Supplementary Figs. S40-41). A rectangular lattice with a main-chain periodicity of 1.30 nm and inter-chain separation of 1.36 nm was clearly revealed (Fig. 5e, f), consistent with the PXRD result of CP2-*s*. Furthermore, an additional inter-chain separation of 1.25 nm was observed within the same fiber (Supplementary Fig. S42). This observation suggests the co-existence of different molecular orientations within the same CP2-*s* nano-crystal, similar to the findings observed in CP1-*s*. Overall, these cryo-EM results elucidated structural details of non-diacetylene-based topochemical polymers from

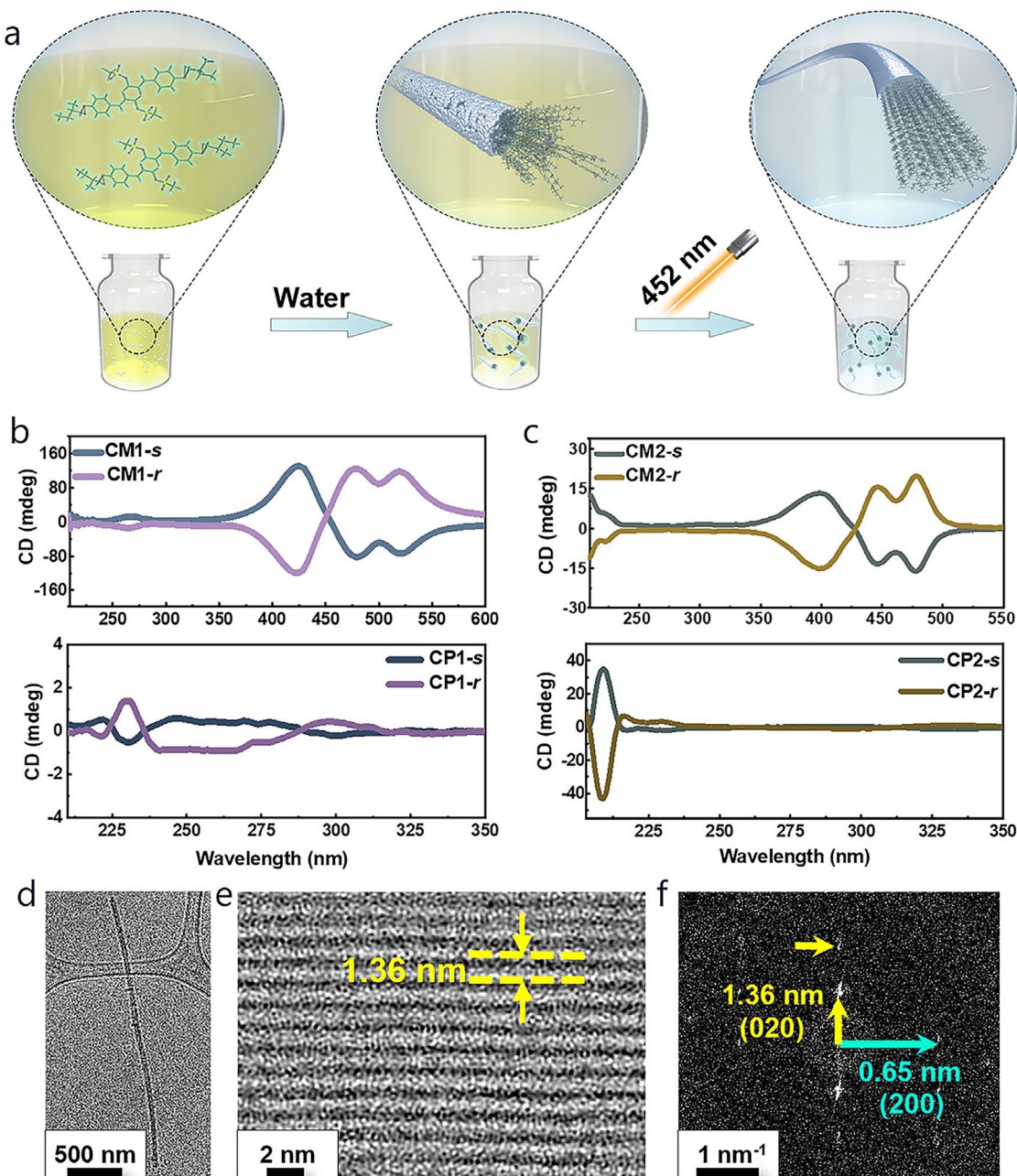

**Fig. 5 | TCP of AQMs in aggregates. a** Illustration of the formation of crystalline polymer fibers via antisolvent-driven self-assembly and subsequent polymerization in aggregates. **b** CD spectra showing the mirror responses of CM1-*s*/CM1-*r*, and CP1-*s*/CP1-*r*. **c** CD spectra showing the mirror responses of CM2-*s*/CM2-*r*, and CP2-*s*/CP2-*r*. **d** Low magnification cryo-EM image of CP2-*s* nanofibers obtained by drop casting the mixture from THF/water. **e** High-resolution cryo-EM images of CP2-*s* nanofibers. **f** Corresponding FFT image, indicating the main-chain periodicity of 1.30 nm and the inter-chain separation of 1.36 nm.

antisolvent-induced supramolecular aggregates, demonstrating a striking similarity as those polymers obtained from solid state polymerization.

In summary, the TCP processes of chiral AQM molecules were elucidated in single crystals, polycrystalline states, and supramolecular aggregates in a liquid medium, respectively. In situ SCXRD analysis of chiral CM1-*s* reveals a series of X-ray-induced SCSC transformations, including a rare metastable phase transition and subsequent TCP. The metastable phase preconditions the crystals such that lattice strains are effectively dissipated during the TCP reaction, successfully retaining the integrity of the final polymer single crystal. Detailed in situ studies of reactions in powders and thin films reveal side-chain

dependent fast kinetics (up to 0.19 s$^{-1}$) and low activation energies (up to 5.9 kcal mol$^{-1}$) for the solid-state reactions. Furthermore, TCP of these chiral AQM molecules has been successfully implemented in a liquid medium by simply manipulating the solvent environment, generating highly crystalline colloidal polymer nanofibers. The nanofibers preserve similar structural details as seen in polymers obtained from solid state reactions, as confirmed by high-resolution Cryo-EM studies. The demonstrated approach to ordered polymer nanofibers combines the high dispersibility inherent to supramolecular polymers and the crystallinity of TCPs, thus opening the door to the fabrication of processable nanostructured polymers with extraordinary structural integrity and desirable morphologies.

## Methods

### Preparation of monomer single crystals

The CM1-*s/r* and CM2-*s/r* were synthesized according to previous reported methods (Supplementary information). All monomer powder were firstly dissolved in organic solvent ($CHCl_3$) and filtered through syringe filters into vials that were shielded from light by wrapping them with aluminum foil. The vial caps were slightly unscrewed to allow slow evaporation of the solvent at 5 °C. Long needle-like single crystals of CM1-*s*, CM1-*r*, CM2-*s* and CM2-*r* were obtained within one week. These crystalline monomers were used for polymerization after vacuum drying under nitrogen.

### Preparation of CP1-*s* polymer single crystal via in situ X-ray irradiation

A single crystal of **CM1-*s*** was mounted on the diffractometer at beamline 12.2.1 at Advanced Light Source (ALS), Lawrence Berkeley National Laboratory (LBNL). The structure of the monomer single crystal was first measured, then the crystal exposed to X-rays at 200 K until the orangish color faded (around 4 h). The structural data of the obtained colorless crystal was collected afterwards.

### Preparation of CP1-*s/r* and CP2-*s/r* polycrystals

**Polymerization by X-ray.** The poly-crystalline CM1-*s*, CM1-*r*, CM2-*s* and CM2-*r* polymers were obtained by recrystallization from $CHCl_3$. These monomer crystals were subjected to continuous X-ray irradiation during the PXRD data acquisition (~10 h; each round takes ~ 2 h), resulting in the formation of polycrystalline CP1-*s/r* and CP2-*s/r* materials.

**Polymerization by heat.** The poly-crystalline monomers were put in a vial purged with $N_2$ and heated at 100 °C for 1 h, after which the colorless polycrystalline CP1-*s/r* and CP2-*s/r* were obtained.

**Polymerization by 452 nm light.** The polymer crystals were obtained after subjecting the monomers to 452 nm light in a UV reactor for 15–30 min.

### Preparation of CP1-*s/r* and CP2-*s/r* thin films

Monomer solutions of CM1-*s/r* and CM2-*s/r* were prepared in $CHCl_3$ at a concentration of 8.4 mM. Under $N_2$ atmosphere, 50 μL of each prepared solution was deposited onto an ITO glass (1.0 × 1.5 cm). These substrates were then subjected to spin coating at 1000 rpm for 1.0 min.

### Preparation of CP1-*s/r* and CP2-*s/r* in dilute solutions

Monomer solutions of CM1-*s/r* and CM2-*s/r* were prepared in $THF/H_2O$ mixtures with $H_2O$ content ranging from 0 to 80% (total concentration of 14 μM). The resulting aggregates dispersed in transparent solution were irradiated with 452 nm light for 15 min. This process caused the yellowish solution to become colorless, indicating the completion of TCP process.

### In situ GIWAXS measurements

Grazing-incidence wide-angle X-ray scattering (GIWAXS) was performed during spin coating and thermal annealing in a custom-made spin coater attached to beamline 12.3.2 at the ALS, LBNL. The incoming X-ray beam was at a shallow angle of 1° with a beam energy of 10 keV. A DECTRIS Pilatus 1 M X-ray detector at an angle of 40° to the sample plane and a sample–detector distance of ~164 mm was used. The diffraction data was collected with a frame rate of ~1.875 s$^{-1}$.

### In situ UV-vis analysis

UV-vis measurements were performed using a home-made device equipped with tungsten-halogen and deuterium light sources over the wavelength range of 350–600 nm[46]. The power density of the light source was measured to be ~131 μW. Transmission measurements were collected using a fiber-coupled Ocean Optics spectrometer (Flame) with an integration time of 0.1 s per transmission spectrum. The equation $[A_\lambda = -\log_{10}(T_\lambda)]$ was used to calculate the UV-visible absorption spectra from the transmission spectra, where $A_\lambda$ is the absorbance at a certain wavelength ($\lambda$) and $T_\lambda$ is the corresponding transmitted radiation. The extent of polymerization was determined by measuring the change in absorption intensity of the monomers at 452 nm.

### Cryo-EM analysis

A 4 μL aliquot of sample solution was placed on a glow-discharged, ultra-thin lacey carbon film 200 mesh gold grid (LC200-Au-UL, Electron Microscopy Sciences, Hatfield, PA, USA). The grid was plunge-frozen in liquid ethane at ~ 90% humidity and 8 °C using a Leica EM GP rapid-plunging device (Leica, Buffalo Grove, IL, USA) after being blotted for 3.5 s with filter paper. The flash-frozen grids were transferred into liquid nitrogen for cryo-EM Imaging. The grid was imaged on a Titan Krios G3i TEM (Thermo Fisher Scientific) equipped with a Bio Quantum energy filter (Gatan Inc., Pleasanton, CA, USA) and operated at 300 keV. Micrographs were acquired using a Gatan K3 direct electron detector in correlated double sampling (CDS) mode[47] and super-resolution mode, controlled by SerialEM[48]. The micrographs were acquired at a nominal magnification of 215 k× (0.41 Å/pixel) or 165 k× (0.52 Å/pixel) with 50 frames. The total dose was ~50 e$^-$/Å$^2$. Motion correction of the multi-frame was conducted by MotionCor2[49]. The Contrast Transfer Function (CTF) was determined and phase flipped using the GCTF software package[50]. Micrographs were analyzed by Fast Fourier Transform (FFT) and processed by Wiener filter[51] and low-pass filter (2 ~ 3 Å) in Gatan Digital Micrograph software.

## Data availability

Crystallographic data of small molecules and polymers are available from Cambridge Structural Database with the following deposition numbers: CM1-*s*, 2359881; CM1-*r*, 2359880; CM2-*s*, 2359878; CM2-*r*, 2359879; CM1-*s\**, 2359882; CP1-*s*, 2359883. The movie related to SCSC process in this study is provided in supplementary movie 1. All data are available from the corresponding author upon request.

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

## Acknowledgements

Work at the Molecular Foundry was supported by the Office of Science, Office of Basic Energy Sciences, of the U.S. Department of Energy under Contract No. DE-AC02-05CH11231. Y.L. and H.L. acknowledges the support from the U.S. Department of Energy, Office of Science, Office of Basic Energy Sciences, Materials Sciences and Engineering Division, under Contract No. DE-AC02-05CH11231 within the Inorganic/Organic Nanocomposites Program (KC3104). Y.L. and C.Y. acknowledge the support from the Laboratory Directed Research and Development (LDRD) Program. GIWAXS was carried out at BL12.3.2 at Advanced Light Source, a user facility supported by the Office of Science, Office of Basic Energy Sciences, of the U.S. Department of Energy under Contract No. DE-AC02-05CH11231. J.L and G.R. were partially supported of the funds from the US National Institutes of Health grants (R01HL115153, R01GM104427, R01MH077303 and R01DK042667). Y.W. and X.G. thank the Department of Energy for funding support under award number DE-SC0022050, which enabled the morphology characterization performed in this work. The authors thank Kaiyue Jiang and Prof. Xiaodong Zhuang for the discussion involving the theoretical structures, Dr. Ziman Chen and Dr. Sizhuo Yang for discussions of the optical properties, Dr. Bezhad Rad for helping with the CD measurements, and Dr. Linfeng Chen for helping with DLS measurements.

## Author contributions

Y.L. and C.Y. designed the experiment and wrote the manuscript; C.Y. conducted the experiment; R.S.H.K. and J.Z. carried out the in situ single crystal diffraction studies; M.A., C.Z. and C.M.S.-F. contributed to the in-situ GIWAXS and transmission analysis; J.L. and G.R. carried out the cryo-EM analysis; X.L. and G.C. carried out calculations; M.Q., H.L., S.H., Q.Z. and L.M.K. helped with experiments and data analysis; H.M. and J.A.R. helped with the SSNMR tests; Y.W., B.Y. and X.G. helped with the AFM tests and analysis. Q.X. developed the tool to analyze the in situ transmission data; All authors contributed to the editing of the manuscript.

## Competing interests

The authors declare no competing interests.
