## [Peer Review file · Nature Communications]

High-Fidelity Topochemical Polymerization in Single Crystals, Polycrystals, and Solution Aggregates

Corresponding Author: Dr Yi Liu

Version 0:

Reviewer comments:

Reviewer #1

(Remarks to the Author)

In this manuscript the authors describe the topochemical polymerization of two azaquinodimethanes derivatives in crystals, thin films and supramolecular aggregate states. In one case the polymerization happens in a SCSC manner when irradiated with X-ray. I don't agree with the solution-phase topochemical polymerization which is highlighted in the manuscript. Topochemical polymerization is impossible in an isotropic solution. What authors see is the reaction in suspended microparticles (solids) in the solvent. This cannot be considered as reaction in solution. I find this study a routine investigation of topochemical reactivity of quinodimethanes. Some of the SCSC topochemical polymerization of quinodimethanes reported in the literature are not cited in the manuscript (<https://doi.org/10.1021/acs.macromol.5b01078>, <https://doi.org/10.1021/acs.macromol.2c00437>). I do not recommend publication of this manuscript in nature communication as it lacks novelty and systematic investigation. Many statements are vague and meaningless. Some specific comments are as follows;

1. The statement "Topochemical polymerization (TCP) emerges as an encouraging protocol enabling the production of macroscopically ordered polymers, via a lattice-controlled single-crystal-to-single-crystal (SCSC) approach under external stimuli" is wrong. Topochemical polymerization need not be SCSC
2. The statement "... potentially changing the distances between monomer reactive groups to a topotactic geometry either facilitating or hindering their original TCP reactivity" is vague and meaningless.
3. The statement "Additionally, significant strain can accumulate within lattice during the SCSC transformation process, easily causing the disintegration of crystals, especially in systems involving large lattice deformation or fast reaction kinetics" is confusing. If the strain is causing disintegration of crystals, it cannot be called as SCSC.
4. Vague phrases such as "topochemical monomers", "...emerging class of TCP-active compounds that readily pack within stereochemical vicinity in the solid state", "distinct xylyl peaks" are plenty in this manuscript.
5. The authors mention that topochemical polymerization is restricted to solid-state transformations and achieving solution-phase TCP is a challenge. First of all, the phrase 'solution-phase TCP' is misleading.
6. What does the authors mean by 'mild TCP reaction'?
7. Chemical reactions between dissolved reactants are regarded as solution-phase reactions. The polymerization described here, in essence happens in the supramolecular aggregate state, formed via phase segregation and crystallization in the presence of water. I am afraid this can not be termed as solution-phase topochemical polymerization.
8. It is mentioned that heat induced polymerization of both the monomers were carried out at 100 °C, which is much higher than the onset temperature for the exothermic peak in the DSC. Authors should try controlled heating, which may preserve the crystal after polymerization. Authors should add temperature dependent PXRD, FT-IR and absorption studies.
9. The DSC profile suggests that the reaction happens at much lower temp than 100 deg C. It is possible that monomer reacts spontaneously (thermal reaction) at room temperature. Is it really an X-ray induced reaction? It may be just a thermal reaction happening at low temperature! The monomer may be reacting thermally during the time taken for X-ray irradiation. A control experiment by keeping the monomer crystals at rt for the duration it was kept in X-ray has to be investigated to confirm or rule out the role of X-ray!
10. Why is there large shift in the quinoidal C=C stretching peaks of CM1 and CM2?
11. Resolution of Fig. 2a should be improved.
12. The authors state that the intermediate structure CM1* was obtained during initial stage of X-ray irradiation and prolonged X-ray exposure leads to polymerization. Phrases like these are subjective. Optimized experimental conditions are to be detailed in the main manuscript.

13. In the DSC of CM1, why is there a sharp endothermic peak at the end of the first and second heating cycle and the exothermic at 157 °C during the cooling cycle? These phase transitions should be investigated.
14. In time-dependent PXRD (Figs. 3a, 3b), the time interval between each measurement should be mentioned for a clear understanding.
15. The authors propose a heterogenous polymer growth mechanism. However, the experiments are not adequate to support this. PXRD is not conclusive. More experiments are to be added.
16. Also, the crystal expands/bends/splits during the polymerization (Video). How does the polymer growth mechanism corroborate with the mechanical responses of the crystal? The heterogenous polymer growth mechanism is contradictory here.
17. Is it possible to do post polymerization functionalization of the polymer by displacing the triflate groups?
18. Authors should take CD in the solid-state and compare with the CD spectra of the suspension to make sure that monomer self-assembly in the aggregate state is same as in the crystal. Any orientational effects also should be ruled out. In short this is a routine paper, with many vague claims without proper proofs. This paper is not suitable for Nature Communications.

Reviewer #2

(Remarks to the Author)

This manuscript provides an in-depth elucidation of the topochemical polymerization processes through the study of chiral AQM molecules across single crystals, polycrystalline states, and the solution state. Utilizing in situ SCXRD analysis, the authors uncovered a series of X-ray-induced SCSC transformations, including a rare metastable phase that mitigates lattice strains during TCP, which overcame the challenge of disintegration of the final polymer single crystals and allowed structural determination by SCXRD. The authors further carried out complementary in situ studies on powders and thin films to reveal side-chain dependent fast kinetics and low activation energies for the solid-state reactions. Moreover, the authors demonstrated the extension of TCP to the solution phase through solvent manipulation, yielding highly crystalline colloidal polymer nanofibers as confirmed by high-resolution Cryo-EM. This work is among the most comprehensive studies of topochemical polymerization processes, offering deep insight with the necessary breath into a highly interesting field of crystalline extended solids. I thus recommend its acceptance for publication with minor concerns to be addressed:

1. Given the solvent-induced assembly behavior of the AQM monomers, how does that impact the emission properties of these monomers and the resulting polymers?
2. What is the peak at ~180 C in the DSC of CM1-s? Does it correspond to polymer melting or phase transition?
3. In Fig. S27 and S28, why did the monomer absorption decrease beyond 75% water content?
4. Were the cryo-TEM studies conducted on fibers in dry state or frozen in ice?
5. Can the authors add a scale bar to Figures 2b-c?
6. There are also some formatting issues that need to be addressed:
 - a. In Figures 3c-d, the temperature axis is not fully displayed, showing ... 50, 100, 150, 20 °C...;
 - b. Reference 24 is incorrectly formatted with two instances of the numbers 24 at the beginning.

Reviewer #3

(Remarks to the Author)

The manuscript describes a well characterized SCSC reaction of involving a relatively unique molecular system. The thrust of the manuscript lies in the idea of transforming the topochemical effect to solution. The authors show that by adjusting water and organic solvent conditions, it is possible to obtain a similar outcome in solution.

While the work is performed at the high level, and the results are intriguing, it is difficult to see why the paper should be published in Nature Comm. The idea of using solvent to affect reaction outcome is an idea that originates in the 1970's and 1980's (e.g., Whitten) on the study of photochemistry. The approach is also used in thin film chemistry and micelle systems.

Admittedly, here there is more of a direct connection to crystalline materials, and recent work by Sureshan speaks to the importance of polymers, but I do not see how the idea is more enabling given literature precedents.

It would seem that the work is more suitable given the high quality for articles in Angew. Chem. or J. Am. Chem. Soc.

Version 1:

Reviewer comments:

Reviewer #1

(Remarks to the Author)

I have gone through this revised manuscript with interest. This manuscript is not suitable for Nature Communications for the following reasons.

The claims raised in this manuscript is fundamentally wrong! Topochemical reactions cannot happen in dissolved state. The very definition of topochemical reactions is the reaction due to constraints offered by the crystal lattice. Authors bring the argument of 'dissolved supramolecular polymers'. I agree self-assembly can lead to non-covalently connected oligomeric species in solution. But they are in dynamic equilibrium with the free monomer units dissolved in the medium. Their exchange frequency is quite high. In other words, these self-assembled species will not have the kind of constraints offered by the 3D crystal lattice. Even if reactions happen in such self-assembled species, such reactions cannot be termed as topochemical reactions.

The polymerization of diyne, cited by the authors, are not occurring in solution but in gels. Gel is not a solution, but a colloidal state where large amount of solvent molecules are entrapped in solid (usually fibrillar) network formed by the gelator (monomer). This is the case even in transparent gels. Topochemical reaction happens in such solid fibers. Additionally, the facts described under the heading 'Aggregation-reinforced TCP in solution' and experimental results obtained clearly establish that the reaction is not happening in 'dissolved supramolecular polymers'! The monomer is completely soluble in THF. But when water as an antisolvent is added, aggregation occurs. Authors have nicely shown this schematically in Fig 5a. Also, such water-added solution (rather suspension) exhibited Tyndall effect showing the scattering of light by the solid particles in the suspension! How then authors can say the reaction is in dissolved supramolecular polymers?

Regarding the crystal-state reaction:

At 333K, the reaction was fast but in the absence of light such increased reactivity was not observed. Authors ruled out any thermal reaction based on this. However, the same authors reported thermal polymerization of very similar monomers (Nature Communications 2021, 12, 6818) differing only in the peripheral substitution of the benzene rings. Additionally, the new experimental temp-dependent PXRD data provided in response to my comments clearly show thermal reaction! The work of Sureshan et al showed that many thermal topochemical polymerization reactions can occur spontaneously at room temperature. I think a more systematic study is required to rule out the thermal reaction.

What are the melting points of the monomers?

Authors have clarified many vague statements, done additional experiments and provided explanations. While some of these explanations are satisfactory, I take serious note of the following:

Authors replied to my general comment that "We emphasize that TCP in our system occurs in supramolecularly aggregated monomers, which form a homogeneous solution under the conditions employed. This differs fundamentally from reactions in suspended microparticles (solids), which involve inhomogeneous systems". Then how the system is showing scattering?!

Authors replied to my concern about novelty that "AQMs exhibit unique electronic properties and structural characteristics, distinguishing their self-assembly and polymerization behavior from that of traditional quinodimethanes. This distinction is critical for understanding their topochemical reactivity and is a key novelty of our work". How this paper is different from the paper published by the same authors (Nature Communications 2021, 12, 6818)?

I disagree with authors response to my comment 7. Authors are patently wrong in calling their reaction as solution-phase topochemical polymerization. The example cited by the authors in support of their argument (Angew. Chem. Int. Ed.2006, 45, 5383-5386) describes topochemical polymerization of diynes in solid microfibers formed via self-assembly (gelation). That is a solid-state reaction. Also authors own scattering studies and schematic representation also contradict their argument.

In response to my comment 8, authors provided Temp-dependent PXRD data. This clearly shows its thermal polymerization reaction at 65 degree C . Even at lower temperature the shoulder peak, presumably due to polymer phase, is seen. This contradicts with authors contention that thermal polymerization does not occur at 333K (60 degree C).

Regarding my comment 9, authors did not explain how X-ray is facilitating the reaction.

Overall the claims made in this manuscript are not substantiated and hence this manuscript is not publishable.

Reviewer #2

(Remarks to the Author)

The author has addressed my concerns, and this article can now be published.

Reviewer #4

(Remarks to the Author)

[Note from the Editor: Reviewer #4 was invited to assess the response given to reviewer #1, especially the claim of topochemical polymerization in the solution state.]

I will express my opinion below regarding reviewer #1's most serious points (second round).

(1) Regarding the phrase "topochemical polymerization in solution" (General Comment and comment 7):

I agree with reviewer #1's opinion that "topochemical polymerization in solution" will cause a serious misleading. As described in this manuscript and indicated by reviewer #1, topochemical reaction including polymerization proceeds in the solid state. Obviously, topochemical polymerization of monomers molecularly dissolved in solution is impossible. The

polymerizations described here are the results of the polymerization in the solid-state of supramolecular aggregates of AQM monomers in a specific solvent mixture (THF and water) as schematically shown in Fig. 5a, and the mixture happened to be transparent, so the authors thought it was a homogeneous solution (it is not). Most importantly, topochemical polymerization occurs in the solid, not in solution. To avoid misleading, the authors should revise “the solution phase”, “solution-phase polymerization”, “solution-based TCP”, and etc. in the manuscript. For example, “topochemical polymerization in the solid-state of supramolecular aggregates dispersed in (transparent) solution” seems to be appropriate.

It seems worthwhile to perform experiments to measure the light scattering of "supramolecular aggregates dispersed in a (transparent) solution" and to estimate the size of the aggregates.

On the other hand, the most important conclusion of this paper seems to be that it has been discovered that crystalline polymers equivalent to those obtained by topochemical polymerization of single crystals and thin films can be obtained by preparing supramolecular aggregates by adding a poor solvent (water) to a monomer solution in THF and then polymerizing it. Considering the ripple effects in the future, it is felt that there is value in publishing it in this journal after revision according to the above comments.

(2) Regarding the topochemical polymerization mechanism (comments 8 and 9):

To dispel the doubts and concerns that reviewer #1 has about the topochemical polymerization mechanism, the authors should provide more convincing evidence that it is not thermal polymerization.

Version 2:

Reviewer comments:

Reviewer #4

(Remarks to the Author)

The authors performed additional measurements such as DLS and revised the manuscript and Supporting Information based on the new results, which certainly improved the quality of this manuscript. Most of my concerns were also addressed by the authors.

However, the misleading phrases, such as “solution-based TCP” (line 77), “aqueous solutions” (line 87), “TCP of AQMs in solution” (line 385), “in liquid” (line 386), “The resulting solutions” (lines 467-468), and “solution-mediated TCP” (lines 469-470) etc in the revised manuscript have not been sufficiently corrected. There may be other misleading phrases in the revised manuscript and Supporting Information. The authors should carefully correct and/or rephrase as suggested before such as “Most importantly, topochemical polymerization occurs in the solid, not in solution. To avoid misleading, the authors should revise “the solution phase”, “solution-phase polymerization”, “solution-based TCP”, and etc. in the manuscript. For example, “topochemical polymerization in the solid-state of supramolecular aggregates dispersed in (transparent) solution” seems to be appropriate.”

This reviewer may then recommend publication.

REVIEWER COMMENTS

Reviewer #1:

General Comment: In this manuscript the authors describe the topochemical polymerization of two azaquinodimethanes derivatives in crystals, thin films and supramolecular aggregate states. In one case the polymerization happens in a SCSC manner when irradiated with X-ray. I don't agree with the solution-phase topochemical polymerization which is highlighted in the manuscript. Topochemical polymerization is impossible in an isotropic solution. What authors see is the reaction in suspended microparticles (solids) in the solvent. This cannot be considered as reaction in solution. I find this study a routine investigation of topochemical reactivity of quinodimethanes. Some of the SCSC topochemical polymerization of quinodimethanes reported in the literature are not cited in the manuscript (<https://doi.org/10.1021/acs.macromol.5b01078>, <https://doi.org/10.1021/acs.macromol.2c00437>). I do not recommend publication of this manuscript in nature communication as it lacks novelty and systematic investigation. Many statements are vague and meaningless. Some specific comments are as follows;

Response: We appreciate the Reviewer's thorough and insightful feedback. We understand the Reviewer's concern regarding the use of the term "solution-phase TCP" and agree that it requires clarification. We emphasize that TCP in our system occurs in supramolecularly aggregated monomers, which form a homogeneous solution under the conditions employed. This differs fundamentally from reactions in suspended microparticles (solids), which involve inhomogeneous systems. The supramolecular aggregates in our study remain in solution until anti-solvent content becomes excessively high (>80%). This behavior aligns with definitions of "dissolved supramolecular polymers" frequently described in the literature (e.g. Meijer and coworkers, *Nature*, **626**, 1011-1018 (2024)).

We respectfully disagree with the Reviewer's assertion that our study represents a routine investigation of quinodimethane reactivity. AQMs exhibit unique electronic properties and structural characteristics, distinguishing their self-assembly and polymerization behavior from that of traditional quinodimethanes. This distinction is critical for understanding their topochemical reactivity and is a key novelty of our work.

We appreciate the Reviewer pointing out references we inadvertently omitted, and we apologize for this oversight. The cited works (<https://doi.org/10.1021/acs.macromol.5b01078> and <https://doi.org/10.1021/acs.macromol.2c00437>) are now appropriately cited in the revised manuscript (Ref. 14 and 15).

Below, we address each concern in detail and clarify points raised, supported with additional experiments when applicable. We have also made significant revisions to improve the clarity and precision of the manuscript. We hope the updated version and our responses satisfactorily address the concerns raised.

Comment 1. The statement “Topochemical polymerization (TCP) emerges as an encouraging protocol enabling the production of macroscopically ordered polymers, via a lattice-controlled single-crystal-to-single-crystal (SCSC) approach under external stimuli” is wrong. Topochemical polymerization need not be SCSC.

Response: We thank the Reviewer for raising this technical comment. We agree with the Reviewer that topochemical polymerization need not to be SCSC, and have revised the text accordingly to avoid this misunderstanding: “Topochemical polymerization (TCP) emerges as an encouraging protocol enabling the production of macroscopically ordered polymers, via a lattice-controlled approach under external stimuli⁵⁻⁸. The prospect of single-crystal-to-single crystal (SCSC) transformation distinguishes TCP as a rare avenue for producing single crystalline polymers, yet achieving a SCSC transformation during solid-state polymerizations remains nontrivial⁹⁻¹³.”.

Comment 2. The statement “..... potentially changing the distances between monomer reactive groups to a topotactic geometry either facilitating or hindering their original TCP reactivity” is vague and meaningless.

Response: We thank the Reviewer for the feedback. The original intent was to highlight that dynamic molecular motions within monomer crystals can significantly alter packing geometries prior to polymerization. This dynamic behavior, which diverges from traditional static views of TCP, has been revised for clarity in the manuscript: “Recent studies suggest that external stimuli can induce/augment molecular motion within lattices²³⁻²⁵, ~~potentially~~ changing the distances

between monomer reactive groups to a topotactic geometry ~~either facilitating or hindering~~ that impacts their ~~original~~ TCP reactivity²⁶.”.

Comment 3. The statement “Additionally, significant strain can accumulate within lattice during the SCSC transformation process, easily causing the disintegration of crystals, especially in systems involving large lattice deformation or fast reaction kinetics” is confusing. If the strain is causing disintegration of crystals, it cannot be called as SCSC.

Response: We thank the Reviewer for the insightful comment. We meant to state that strain accumulation during TCP can fragment large crystals into smaller grains while maintaining single-crystal domains. The boundry between polycrystal and single crystal can be substantially obscure, as discussed in detail by Ivan Halasz (*Cryst. Growth Des.* **10**, 2817-2823 (2010)). Such boundry becomes more ambiguous due to recent technical advances such as cryo-electron microscopy-based micro-electron diffraction, which demonstrates that very small grains can retain single-crystalline integrity. As having been shown by us and by others, careful control of the polymerization process can mitigate the fragmentation of large crystals despite the presence of significant strains due to lattice mismatch between the monomer and the polymer (*Chem. Soc. Rev.*, **50**, 4062-4099 (2021); *Nat. Commun.*, **12**, 6818 (2021)).

To avoid confusion, we have revised the text accordingly: “Additionally, significant strain can accumulate within lattice during the ~~SCSC transformation~~ TCP process, easily causing the disintegration of crystals, especially in systems involving large lattice deformation or fast reaction kinetics^{5,28-29}.”.

Comment 4. Vague phrases such as “topochemical monomers”, “...emerging class of TCP-active compounds that readily pack within stereochemical vicinity in the solid state”, “distinct xylyl peaks” are plenty in this manuscript.

Response: We thank the Reviewer for the feedback. We have made changes accordingly as follows:

- a. Replaced “topochemical monomers” with “monomers”;
- b. Revised “...emerging class of TCP-active compounds that readily pack within stereochemical vicinity in the solid state”, to “Our study is centered around azaquinodimethanes (AQMs)³⁴, a recently recognized class of compounds with thermally and photochemically induced linear

chain polymerization activity³⁵. These molecules exhibit a propensity to pack in close stereochemical proximity in the solid state, facilitating polymerization.”;

c. Revised “distinct xylyl peaks” to “distinct methine peaks”.

Comment 5. The authors mention that topochemical polymerization is restricted to solid-state transformations and achieving solution-phase TCP is a challenge. First of all, the phrase ‘solution-phase TCP’ is misleading.

Response: We acknowledge the Reviewer’s concern and have clarified this point. The term of ‘solution-phase TCP’ refers to a TCP process occurring in a liquid medium, as opposed to TCP reactions that take place in the solid-state. Please refer to our detailed response to Comment 7 of this Reviewer.

Comment 6. What does the authors mean by ‘mild TCP reaction’?

Response: We thank the Reviewer for pointing out the ambiguity. The description ‘mild TCP reaction’ refers to a TCP process that occurs with fewer spatial constraints, compared to TCP reactions in crystalline environments with rigid and highly ordered lattices. We have replaced “mild TCP reaction” with “facile TCP reaction”, and the sentence now reads: “Replicating the additive-free, and ~~mild~~ facile TCP reaction in less confined medium, like solutions, would open up new paradigm for engineering processable, crystalline polymeric materials.”.

Comment 7. Chemical reactions between dissolved reactants are regarded as solution-phase reactions. The polymerization described here, in essence happens in the supramolecular aggregate state, formed via phase segregation and crystallization in the presence of water. I am afraid this can not be termed as solution-phase topochemical polymerization.

Response: We thank the Reviewer for the very insightful comment. We fully agree that the monomer molecules form supramolecular aggregates in solution, which act as the basis for subsequent TCP reaction of monomers in solution, distinguishing it from TCP occurring in solid-state system (*Angew. Chem. Int. Ed.* **45**, 5383-5386 (2006)). Such supramolecular aggregates, frequently referred to as “supramolecular polymers” in the literature, are “known to form homogeneous solutions and gels under dilute and concentrated conditions”, as described by Meijer and coworkers (*Nature*, **626**, 1011-1018 (2024)), justifying them as solution species. Importantly,

these supramolecular aggregates don't phase segregate and remain homogeneous in dilute solution, as is shown in our case (concentration: 14 μM). Thus, we believe the reaction occurs within the solution phase albeit in aggregated monomer units.

Regarding the aspect of TCP, the observed reaction fits the IUPAC definition of TCP, which is “a polymerization process in which the spatial arrangement of the reacting monomers in the crystal lattice dictates the polymerization pathway and structure of the resulting polymer.” In our process, the polymerization occurs in a highly organized and controlled manner (proved by cryo-EM), in a homogeneous solution of supramolecularly pre-organized monomers as opposed to the more conventional solid state.

Comment 8. It is mentioned that heat induced polymerization of both the monomers were carried out at 100 °C, which is much higher than the onset temperature for the exothermic peak in the DSC. Authors should try controlled heating, which may preserve the crystal after polymerization. Authors should add temperature dependent PXRD, FT-IR and absorption studies.

Response: We appreciate the Reviewer for the suggestions. In fact we have tried thermal polymerization which only resulted in lower quality crystals. A high temperature of 100 °C was applied to ensure all the monomers were converted to polymers. We would like to note that, the reason we chose to synthesize polymers under such conditions was to quickly produce quantities large enough for IR and solid-state NMR characterization. These analyses focus on the polymers' chemical properties and are not directly influenced by their crystalline properties.

Figure R1. Variable temperature PXRD of CM1-s.

Following the Reviewer's suggestions, we have performed additional variable-temperature PXRD (VT-PXRD) studies. VT-PXRD data of CM1-s was collected after temperature stabilized at each temperature (30, 54 and 65 °C, as well as after cooling back to 30 °C) for 20 min. As shown in Figure **R1**, no significant changes were observed at 30 and 54 °C. However, at 65 °C, a small side peak emerged at ~6.5°, indicating the occurrence of polymerization despite that it is slightly below DSC polymerization temperature. Interestingly, after cooling back to 30 °C, the monomer peak at 5.4° disappeared, leaving only the polymer peak at 6.5°. Such results indicated that thermal-induced TCP had been initiated when the temperature was raised to ~ 65 °C, and proceeded to complete polymerization even without further increasing the temperature, presumably via an auto-catalytic process. Note that the heating module in our PXRD is subjected to overheating in a range of 2–5 °C, which may account for the difference between the onset polymerization temperatures observed by DSC and VT-PXRD.

Regarding temperature-dependent absorption studies, Figure 4 has already detailed the *in-situ* absorption studies of TCP reactions of two AQM monomers carried out at different temperatures. Unfortunately, we don't have resources to carry out temperature-dependent FT-IR studies. Nonetheless, we believe the absorption studies, together with the VT PXRD data, provide sufficient details regarding the behavior of temperature-dependent TCP.

Comment 9. The DSC profile suggests that the reaction happens at much lower temp than 100 deg C. It is possible that monomer reacts spontaneously (thermal reaction) at room temperature. Is it really an X-ray induced reaction? It may be just a thermal reaction happening at low temperature! The monomer may be reacting thermally during the time taken for X-ray irradiation. A control experiment by keeping the monomer crystals at rt for the duration it was kept in X-ray has to be investigated to confirm or rule out the role of X-ray!

Response: Thanks for the insightful suggestion from this Reviewer. To rule out spontaneous thermal reaction at room temperature, CM1-s powder was stored in the dark for 0, 2, 4, 6 and 8 h at r.t., designated as CM1-s, CM1-1, CM1-2, CM1-3 and CM1-4, respectively (Figure **R2a**). Their PXRD patterns were recorded via fast scan (within 5 min) to minimize X-ray induced effect. As shown in Figure **R2a**, the PXRD patterns stored in the dark remained unchanged over a period of 8 h, in contrast to those exposed to continuous X-ray irradiation (Figure **R2b**). These results confirm

unambiguously that it is X-ray irradiation, rather than heat, that drives the TCP at room temperature.

We have added the following discussion to the main text (Page 12), with the new figure included in Supplementary Information as Fig. S12.: “In the absence of X-ray irradiation, the PXRD pattern remained unchanged, further confirming that X-ray irradiation induced the observed TCP (Supplementary Fig. S12).”

Figure R2. Time-dependent powder XRD spectra of CM1-*s*: (a) CM1-*s* stored in the dark at r.t. without X-ray exposure, measured at time intervals of 0, 2, 4, 6 and 8 h, denoted as CM1-*s*, CM1-1, CM1-2, CM1-3 and CM1-4, respectively. (b) Samples under continuous X-ray exposure. The number 1, 2, 3, 4 indicate cumulative X-ray irradiation durations of 2, 4, 6, and 8 hours, respectively.

Comment 10. Why is there large shift in the quinoidal C=C stretching peaks of CM1 and CM2?

Response: We appreciate the insightful question from this Reviewer. The shift can be understood by the different electronic effect: the end group of CM1 is a *para*-substituted phenoxy group, which facilitates both effects of inductively withdrawing and resonatively donating on the conjugating AQM core, while in the case of CM2, the end group is a 3,5-substituted phenoxy group, which only exerts an inductively withdrawing effect. Such a difference accounts for shift observed in the FT-IR spectra.

Comment 11. Resolution of Fig. 2a should be improved.

Response: We thank the Reviewer for pointing this out. As suggested, the resolution of Fig. 2a has been increased to 600 dpi.

Comment 12. The authors state that the intermediate structure CM1* was obtained during initial stage of X-ray irradiation and prolonged X-ray exposure leads to polymerization. Phrases like these are subjective. Optimized experimental conditions are to be detailed in the main manuscript.

Response: Thanks for the kind reminder. We have modified the related text as follows: “Remarkably, a metastable intermediate crystal phase (denoted as CM1*-s) was observed during ~ 3 hours of X-ray irradiation at 253 K, which was followed by gradual transformation to the final polymer crystal phase within 4 hours (Figs. 2d-e), as elucidated by *in situ* X-ray structure analysis.”.

Comment 13. In the DSC of CM1, why is there a sharp endothermic peak at the end of the first and second heating cycle and the exothermic at 157 °C during the cooling cycle? These phase transitions should be investigated.

Response: Thanks for the insightful question. As shown by the DSC data in Figure R3, the TCP completed at above 100 °C. Therefore, the sharp endothermic peaks observed in the first and second cycles at around 180 °C could be assigned to the polymer phase change behavior. As shown in Figure R3, after polymerization, CP1-s exhibited a repeatable endothermic peak at ~180 °C, along with a broad exothermic peak at ~157 °C upon cooling. These phenomena correspond to a reversible thermally induced phase change in the polymer crystals.

Figure R3. DSC curves of (a) CM1-s monomer and (b) CP1-s polymer.

Variable-temperature PXRD analysis of CP1-*s* was also conducted to provide further evidence of the phase transition. As illustrated in Figure R4, a new side peak centered at 5.7° started to emerge at 190 °C, whereas only a sharp peak at 6.5° was observed at 175 °C. Upon cooling back to room temperature, the new peak at 5.7° disappeared. This result confirms a reversible phase transition induced by heat, which is tentatively attributed to thermally driven reorganization of side chains in the polymer.

Figure R4. Variable-temperature PXRD of CP1-*s* showing a reversible phase change at high temperatures.

We have included a discussion of such polymer phase change in our revised manuscript (Page 13) and supplementary information (Supplementary Figs. S17-18): “The additional sharp endothermic peaks observed at ~180 °C corresponded to a reversible crystal-to-crystal phase change of the polymer, as was corroborated by variable temperature PXRD studies of CP1-*s* (Supplementary Figs. S17-18).”

Comment 14. In time-dependent PXRD (Figs. 3a, 3b), the time interval between each measurement should be mentioned for a clear understanding.

Response: Thanks for the kind suggestion. The time interval between each measurement is ~2.0 h. We have included this information in the caption of Figure 3a-b as follows: “The numbers 1, 2, 3, 4 and 5 correspond to cumulative X-ray irradiation durations of 2, 4, 6, 8 and 10 hours, respectively.”

Comment 15. The authors propose a heterogenous polymer growth mechanism. However, the experiments are not adequate to support this. PXRD is not conclusive. More experiments are to be

added. Also, the crystal expands/bends/splits during the polymerization (Video). How does the polymer growth mechanism corroborate with the mechanical responses of the crystal? The heterogeneous polymer growth mechanism is contradictory here.

Response: We thank the Reviewer for the insightful comments. We recognize the importance of providing robust evidence for the proposed heterogeneous polymer growth mechanism and have expanded the discussion and included additional data to clarify this point.

- a. In typical heterogeneous polymerization, the process initiates preferentially at defect sites in the mother crystal and progresses on the surfaces of pre-existing nuclei, leading to fragmentation of the parent crystal into polycrystalline aggregates as the reaction proceeds (*J. Polym. Sci., Polym. Phys. Ed.* **16**, 1365-1378 (1978); *Chem. Sci.*, **3**, 2301-2306 (2012)). Therefore, heterogeneous TCP is usually characterized by the disintegration of the parent crystal, as observed in the diacetylene systems (*Chem. Phys. Lett.* **71**, 44-48 (1980); *Acc. Chem. Res.* **41**, 1215-1229 (2008)). In contrast, during homogeneous polymerization, polymer chains grow randomly and are uniformly distributed within the parent crystal, forming a solid solution (*J. Am. Chem. Soc.* **104**, 6556-6561 (1982)).
- b. In addition to PXRD analysis, the proposed heterogeneous mechanism of the AQM system is further supported by the *in-situ* video observations. As shown in Figure **R5**, polymerization was initiated in localized regions—primarily at the edges and defects of the single crystal—causing these regions to turn colorless within ~2 minutes of light exposure. These localized polymerization zones expanded over time, accompanied by visible crystal splitting. This observation strongly supported a heterogeneous mechanism for the AQM system.

Figure R5. Snapshots of the CM1-*s* crystal under continuous exposure to ambient light at different time intervals (the unit is in minutes), showing the heterogeneous polymerization that led to crystal disintegration.

- c. We hold a different perspective regarding the comment “The heterogeneous polymer growth mechanism is contradictory here”. During polymerization, crystal deformation (expansion/bending/splitting) generates additional defect sites, facilitating further polymer growth. This self-reinforcing behavior accelerates the reaction rate, which is further evidenced by kinetic studies (Figures S24-26).
- d. The mechanical responses and resultant polydispersity of the polymer pose significant challenges for characterizing the single-crystal to single-crystal transformation via SCXRD (*Chem. Sci.*, **3**, 2301-2306 (2012); *Nat. Commun.*, **12**, 6818 (2021)). Therefore, the SCSC process could only be possible by carefully control the reaction conditions, i.e., at low temperatures under continuous X-ray irradiation, as we have demonstrated here.

We have included the following discussion in the revised main text (Page 12) and Supplementary Information (Figure S14): “This finding was consistent with the *in-situ* video results (Supplementary Fig. S14), which revealed that polymerization was initiated in localized regions—primarily at the edges and defects of the single crystal—causing these regions to turn colorless within ~2 minutes of light exposure. These localized polymerization zones expanded over time,

accompanied by visible crystal splitting. This observation strongly supported a heterogeneous mechanism for the AQM system.”

Comment 16. Is it possible to do post polymerization functionalization of the polymer by displacing the triflate groups?

Response: Thanks for the thoughtful question. We have previously observed that the triflate groups in AQM-derived TCP polymers could be displaced by post polymerization functionalization of polymers with good solubility (*Nat. Commun.*, **12**, 6818 (2021)). Preliminary results indicate that insoluble TCP polymers can also undergo reactions with nucleophiles, such as amines, although chain scission was observed. This avenue of research remains highly promising, and detailed studies will be reported in future work.

Comment 17. Authors should take CD in the solid-state and compare with the CD spectra of the suspension to make sure that monomer self-assembly in the aggregate state is same as in the crystal. Any orientational effects also should be ruled out.

Response: We thank the Reviewer for the comment. Following this recommendation, we have attempted to acquire CD spectra of CM1-*s/r* and CP1-*s/r* thin films at room temperature. It was observed, however, the monomer thin films underwent polymerization during CD measurements (~3 minutes) under weak CD light irradiation, consistent with the *in-situ* UV-vis kinetic results (Figure 4). Despite that, CD spectra of the resulting CP1-*s/r* polymer thin films were obtained and compared to that of the polymerized fibers in dilute solutions, which showed great resemblance in the 200–300 nm range. These findings support that the self-organization geometry of AQM monomers remains consistent across the examined states.

Figure R6. CD spectra of CP1-*s/r* in (a) dilute mixed solutions and (b) in thin films.

General Comment: In short this is a routine paper, with many vague claims without proper proofs. This paper is not suitable for Nature Communications.

Response: We respectfully disagree with the Reviewer's assessment of our work as routine and would like to summarize the key contributions that distinguish this study:

- a. We performed challenging *in-situ* single-crystal XRD experiments to elucidate the TCP process. These studies revealed unprecedented details of the phase-change behavior in an AQM monomer, a notable addition to the limited collection of TCP monomeric systems.
- b. We developed a solution-based TCP approach that enables the production of solution-accessible polymer nanostructures approaching single-crystalline quality. This innovation expands the scope of TCP beyond the traditional solid-state domain.
- c. Using cryo-EM, we validated the structural details of the crystalline nanofibril polymers, offering unequivocal evidence of their unique morphology and crystallinity.
- d. Detailed kinetic studies provided novel insights into the TCP mechanism and its structure-property relationships, contributing to a deeper understanding of this process.

We thank the Reviewer for the opportunities to improve the clarity of our manuscript and address their concerns. With these detailed responses and revisions, we believe the work demonstrates sufficient novelty and significance for the high standards of *Nature Communications*.

Reviewer #2 (Remarks to the Author):

General Comment: This manuscript provides an in-depth elucidation of the topochemical polymerization processes through the study of chiral AQM molecules across single crystals, polycrystalline states, and the solution state. Utilizing in situ SCXRD analysis, the authors uncovered a series of X-ray-induced SCSC transformations, including a rare metastable phase that mitigates lattice strains during TCP, which overcame the challenge of disintegration of the final polymer single crystals and allowed structural determination by SCXRD. The authors further carried out complementary in situ studies on powders and thin films to reveal side-chain dependent fast kinetics and low activation energies for the solid-state reactions. Moreover, the authors demonstrated the extension of TCP to the solution phase through solvent manipulation, yielding highly crystalline colloidal polymer nanofibers as confirmed by high-resolution Cryo-EM. This work is among the most comprehensive studies of topochemical polymerization processes, offering deep insight with the necessary breath into a highly interesting field of crystalline extended solids. I thus recommend its acceptance for publication with minor concerns to be addressed:

Response: We sincerely appreciate the encouraging comments from the Reviewer. We have made additional efforts to address the concerns raised, as detailed below.

Comment 1. Given the solvent-induced assembly behavior of the AQM monomers, how does that impact the emission properties of these monomers and the resulting polymers?

Response: We appreciate the insightful question from this Reviewer. Based on the adsorption properties of AQM monomers (Figures **R7a-b**), excitation wavelengths of 460 and 432 nm were selected for CM1-*s* and CM2-*s*, respectively. As shown in Figure **R7c**, when fully dissolved in THF, the excitation spectra of CM1-*s* displayed a weak and broad emission peak centered at ~540 nm. Upon the addition of 75% water, a more distinct fluorescence peak appeared at ~600 nm. This observation indicated an aggregation induced emission effect (*Chem. Rev.* **115**, 11718-11940 (2015); *J. Am. Chem. Soc.* **141**, 8412-8415 (2019)) associated with the AQM cores. For CM2-*s*, the fully dissolved form exhibited a very broad and weak emission peak around 625 nm (Figure **R7d**). Upon water-induced aggregation, the emission was further quenched, suggesting that CM2-*s* experienced an aggregation-quenching effect in the mixed solvent system.

Figure R7. The absorption spectra of CM1-s (a) and CM2-s (b) in pure THF and mixed solvent (H₂O/THF; 75%/25%); The fluorescence spectra of CM1-s (c) and CM2-s (d) in pure THF and mixed solvent (H₂O/THF; 75%/25%).

After polymerization, no fluorescence signals were detected in solution when excited at 360 nm. We also checked the fluorescence properties of powder samples prepared from solid state TCP. As shown in Figure R8, no emission peaks were detected for CP1-s, while CP2-s was weakly emissive, with an emission peak centered at 478 nm.

Figure R8. Fluorescence spectra of CP1-s and CP2-s powder crystals prepared by solid state TCP.

Comment 2. What is the peak at ~180 C in the DSC of CM1-s? Does it correspond to polymer melting or phase transition?

Response: Thanks for the insightful question. We attribute this peak to a reversible crystal-to-crystal phase transition associated with polymer CP1-*s*. Please refer to our detailed response to comment 13 of Reviewer 1.

Comment 3. In Fig. S27 and S28, why did the monomer absorption decrease beyond 75% water content?

Response: Thanks for the insightful questions. It is reasoned that H₂O serves as a poor solvent for both CM1 and CM2. Upon the addition of H₂O, both monomers tend to aggregate through π - π stacking. While the H₂O content exceeds 75%, the resulting aggregates may phase separate from the mixed solvent system, leading to the decrease of UV-vis adsorption (*Chem. Rev.* **115**, 11718-11940 (2015); *New J. Chem.*, **41**, 4747-4749 (2017)).

Comment 4. Were the cryo-TEM studies conducted on fibers in dry state or frozen in ice?

Response: We thank the Reviewer for this question. The cryo-TEM studies were conducted on fibers frozen in ice.

Comment 5. Can the authors add a scale bar to Figures 2b-c?

Response: Thanks for this suggestion. As recommended, the scale bar has been added in Figure **R9**.

Figure R9. Updated figure with scalebars inserted.

Comment 6. There are also some formatting issues that need to be addressed:
a. In Figures 3c-d, the temperature axis is not fully displayed, showing ... 50, 100, 150, 20 °C...;
b. Reference 24 is incorrectly formatted with two instances of the numbers 24 at the beginning.

Response: We appreciate the Reviewer for pointing these out. We have addressed these mistakes accordingly in our revised manuscript.

Reviewer #3 (Remarks to the Author)::

General Comment: The manuscript describes a well characterized SCSC reaction of involving a relatively unique molecular system. The thrust of the manuscript lies in the idea of transforming the topochemical effect to solution. The authors show that by adjusting water and organic solvent conditions, it is possible to obtain a similar outcome in solution. While the work is performed at the high level, and the results are intriguing, it is difficult to see why the paper should be published in Nature Comm. The idea of using solvent to affect reaction outcome is an idea that originates in the 1970's and 1980's (e.g., David Whitten) on the study of photochemistry. The approach is also used in thin film chemistry and micelle systems. Admittedly, here there is more of a direct connection to crystalline materials, and recent work by Sureshan speaks to the importance of polymers, but I do not see how the idea is more enabling given literature precedents. It would seem that the work is more suitable given the high quality for articles in Angew. Chem. or J. Am. Chem. Soc.

Response: We sincerely thank the Reviewer for their thoughtful feedback and for highlighting the relevance of early works by David Whitten on photochemistry in confined environments. We agree that the underlying principle of leveraging supramolecular organization is shared, but we believe our work demonstrates critical distinctions and significant advancements in this area. In particular, we would like to emphasize the following points:

- a. Our work involves chain polymerization, which fundamentally differs from the cyclodimerization studied in Whitten's research. This distinction is not just reaction type (polymer vs. small molecules) but also extends to the nature of confinement. Unlike Whitten's systems, which rely on hydrogen bonding or amphiphilic properties, the confinement in our system is achieved through an entirely different mechanism, i.e., the high propensity of π - π stacking of the aromatic AQM core.
- b. Thin-film approaches, while innovative, lack the solution processability we achieve here. On the other hand, micelle systems, although they allow for some degree of supramolecular organization, cannot attain the structural precision demonstrated in our work. The ability to conduct topochemical polymerization in solution remains highly challenging and, to date, is largely limited to diacetylene derivatives. These systems require directing groups for supramolecular organization, and even then, the structural fidelity falls short of the precision achieved in our study.

- c. Beyond the solution TCP, our work offers several unique contributions, including a) *in-situ* single crystal XRD studies, which remain very challenging, provide unprecedented insights into the phase-change behavior during the polymerization of an AQM monomer; b) through detailed kinetic studies (*in-situ UV-vis*), we have uncovered unique aspects of the TCP mechanism and the structure-property relationships that govern this process. These findings offer a deeper understanding of the system that we believe will inform future studies in the field.
- d. Functional polymers have emerged as promising candidates for advanced electronic applications, such as high-mobility transistors and dielectric capacitors. Among these, novel single-crystal polymers, particularly nanostructured ones, are invaluable for elucidating precise structure–property relationships that underpin their exceptional performance. Our previous work have demonstrated that AQM-derived polymers exhibit outstanding dielectric capacitor properties (*Nat. Commun.*, **12**, 6818 (2021)). We envision that such single-crystalline nanomaterials hold significant potential as model systems for advancing our understanding of their structure–property relationships.

We greatly appreciate the Reviewer’s positive remarks about the quality of our work and their acknowledgment of its suitability for leading journals such as *Angew. Chem. Int. Ed.* or *J. Am. Chem. Soc.* We would like to point out that similar topical research has been published in *Nature Communications* by Sureshan (as exemplified by the Reviewer) and coworkers, such as:

- Balan, H., Sureshan, K.M. *Hierarchical single-crystal-to-single-crystal transformations of a monomer to a 1D-polymer and then to a 2D-polymer.* *Nat. Commun.* **15**, 6638 (2024).
- Khazeber, R., Pathak, S., Sureshan, K.M. *Simultaneous and in situ syntheses of an enantiomeric pair of homochiral polymers as their perfect stereocomplex in a crystal.* *Nat. Commun.* **15**, 6639 (2024).
- Mohanrao, R., Hema, K., Sureshan, K.M. *Topochemical synthesis of different polymorphs of polymers as a paradigm for tuning properties of polymers.* *Nat. Commun.* **11**, 865 (2020).

We believe that our findings offer a similarly broad appeal to the audience of *Nature Communications*. In particular, the combination of our solution-based TCP approach, *in-situ* structural studies, and mechanistic insights makes this work a strong fit for the journal’s readership. We hope this clarifies the significance and novelty of our study and its suitability for this journal.

Point-to-point responses

We appreciate the Reviewers' thorough evaluation of our revised manuscript. Below, we address the concerns raised, clarify any misunderstandings, and provide further justification for our claims.

Reviewer #1 (Remarks to the Author):

Comment 1: I have gone through this revised manuscript with interest. This manuscript is not suitable for Nature Communications for the following reasons. The claims raised in this manuscript is fundamentally wrong! Topochemical reactions cannot happen in dissolved state. The very definition of topochemical reactions is the reaction due to constraints offered by the crystal lattice. Authors bring the argument of 'dissolved supramolecular polymers'. I agree self-assembly can lead to non-covalently connected oligomeric species in solution. But they are in dynamic equilibrium with the free monomer units dissolved in the medium. Their exchange frequency is quite high. In other words, these self-assembled species will not have the kind of constraints offered by the 3D crystal lattice. Even if reactions happen in such self-assembled species, such reactions cannot be termed as topochemical reactions.

Response: To avoid the potential confusion regarding “solution-based TCP”, we have revised accordingly to emphasize that such reactions occurred in supramolecular aggregates within a liquid medium. We fully recognize the Reviewer's point about supramolecular polymers existing in dynamic equilibrium in solution, and the concern that reactions in such self-assembled species usually lack the same level of constraints offered by the 3D crystal lattice. It is however noteworthy that our results, as supported by high-resolution cryo-TEM studies, showed that the structures of nanofibril polymers obtained from the liquid medium is nearly identical to those formed via conventional solid-state reactions. Furthermore, as addressed in our response to previous comment 17, the CD spectrum of nanofibril polymers closely resembles that of polymers synthesized through solid-state methods. The structural similarity provides strong evidence that these supramolecular assemblies impose sufficient spatial constraints to enable a topochemical-like polymerization process. Furthermore, our system is distinctive in that it achieves pre-organization and TCP reactivity without relying on strong H-bonding interactions, differentiating it from prior examples in the literature.

Comment 2: The polymerization of diyne, cited by the authors, are not occurring in solution but in gels. Gel is not a solution, but a colloidal state where large amount of solvent molecules are entrapped in solid (usually fibrillar) network formed by the gelator (monomer). This is the case even in transparent gels. Topochemical reaction happens in such solid fibers.

Response: We agree with the Reviewer that gel is distinct from a solution as a fibrillar network that entraps solvent. We'd like to clarify, however, that among the references we cited (ref 33-37), only refs 36 and 37 involved gels, while refs 33-35 described supramolecular polymers formed in solution which were further polymerized without phase separation. These references highlight cases where TCP occurs in pre-assembled structures that do not meet the classical definition of a gel. Note that existing examples are exclusively based on diacetylene monomers. Our study extends these concepts beyond the well-established diacetylene systems, demonstrating that AQM monomers also exhibit versatile TCP behavior. This distinction is a significant contribution of our work.

Comment 3: Additionally, the facts described under the heading 'Aggregation-reinforced TCP in solution' and experimental results obtained clearly establish that the reaction is not happening in 'dissolved supramolecular polymers'! The monomer is completely soluble in THF. But when water as an antisolvent is added, aggregation occurs. Authors have nicely shown this schematically in Fig 5a.

Response: We confirm that the reaction occurs within supramolecular polymers, formed due to antisolvent-induced aggregation, a phenomenon widely observed in supramolecular systems. To prevent any ambiguity on the use of the term "TCP in solution", we have made changes accordingly to indicate that these reactions take place in a liquid medium rather than in a homogeneous solution.

Comments 4: Also, such water-added solution (rather suspension) exhibited Tyndall effect showing the scattering of light by the solid particles in the suspension! How then authors can say the reaction is in dissolved supramolecular polymers?

Response: We believe there is some misunderstanding of the experiments. As shown in Figures S33-34, the yellowish monomer solutions of CM1-s and CM2-s would turn into colorless after polymerization. And the Tyndall effect was exhibited only in the polymerized fibers (in colorless solutions), which formed covalently linked crystalline fibers dispersed in the medium. The supramolecular polymers prior to polymerization do not show similar scattering behavior.

Comments 5: Regarding the crystal-state reaction:

At 333K, the reaction was fast but in the absence of light such increased reactivity was not observed. Authors ruled out any thermal reaction based on this. However, the same authors reported thermal polymerization of very similar monomers (Nature Communications 2021, 12, 6818) differing only in the peripheral substitution of the benzene rings. Additionally, the new experimental temp-dependent PXRD data provided in response to my comments clearly show thermal reaction! The work of Sureshan et al showed that many thermal topochemical polymerization reactions can occur

spontaneously at room temperature. I think a more systematic study is required to rule out the thermal reaction.

Response: We appreciate the opportunity to clarify the point on thermal reactivity. Our statement was not that thermal polymerization is entirely absent, but rather that it occurs at a negligible rate at 333 K in the absence of light. We'd like to clarify that it is not meaningful to rule out spontaneous thermal reaction. As thermodynamic principles dictate, some level of thermal reaction is inevitable at non-zero temperatures. However, it would only become relevant if the thermal dark reaction is fast enough to contribute significantly to the overall reaction rate. Our DSC and in situ PXRD data indicate that the rate of dark polymerization below the DSC polymerization temperature (T_p) is minimal. The lack of conversion in the absence of light, as shown by UV-vis absorption in Figures S27-29 (up to 333 K within 1000 s) and by PXRD in Figure S12a, (8 hrs at RT), has adequately supported our conclusion that the dominant reaction pathway is photoactivated, rather than thermally driven.

Our 2021 study introduced the concept of TCP in AQM derivatives and demonstrated the feasibility of obtaining soluble, ultrahigh-molecular-weight polymers. However, that work did not include detailed structural analyses beyond microED of microcrystals. In contrast, the present study provides a comprehensive investigation of polymerization kinetics and mechanisms using in situ XRD, as well as high-fidelity structural characterization of the polymer products. Additionally, this work explores TCP behavior in both crystalline and self-assembled supramolecular aggregates in a liquid media, significantly expanding our understanding of the topochemical reactivity of AQM monomers. These new insights clearly distinguish this study from our prior publication.

We believe our system is fundamentally different from the main body of Sureshan's work on azide-alkyne addition, which is solely based on ground-state reactivities, while the AQM TCP reactivities can be related to excited-state (photo polymerization) and ground-state (thermal polymerization). We agree that many TCP may occur spontaneously at room temperature. We show through our example that even though there may be thermal reactions, it is slow enough compared to the photo reaction and is thus not relevant to the main points of this work.

Comments 6: What are the melting points of the monomers?

Response: All the monomers undergo polymerization before melting, as evidenced by their DSC profiles. Consequently, the monomers don't show any melting points.

Comments 7: Authors have clarified many vague statements, done additional experiments and provided explanations. While some of these explanations are

satisfactory, I take serious note of the following: Authors replied to my general comment that “We emphasize that TCP in our system occurs in supramolecularly aggregated monomers, which form a homogeneous solution under the conditions employed. This differs fundamentally from reactions in suspended microparticles (solids), which involve inhomogeneous systems”. Then how the system is showing scattering?!

Response: We believe there is some misunderstanding of the experiments. The scattering (Tyndall effect) was exhibited only in the polymerized fibers (in colorless solutions as shown in Figures S33-34), which formed covalently linked crystalline fibers dispersed in the medium. The supramolecular polymers prior to polymerization do not show similar scattering behavior.

Comments 8: Authors replied to my concern about novelty that “AQMs exhibit unique electronic properties and structural characteristics, distinguishing their self-assembly and polymerization behavior from that of traditional quinodimethanes. This distinction is critical for understanding their topochemical reactivity and is a key novelty of our work”. How this paper is different from the paper published by the same authors (Nature Communications 2021, 12, 6818)?

Response: Our 2021 study introduced the concept of TCP in AQM derivatives and demonstrated the feasibility of obtaining soluble, ultrahigh-molecular-weight polymers. However, that work did not include detailed structural analyses beyond microED of microcrystals. In contrast, the present study provides a comprehensive investigation of polymerization kinetics and mechanisms using in situ XRD, as well as high-fidelity structural characterization of the polymer products. Additionally, this work explores TCP behavior in both crystalline and self-assembled supramolecular aggregates in a liquid media, significantly expanding our understanding of the topochemical reactivity of AQM monomers. Details about polymerization in crystals and solution-born self-assembled states, as well reaction kinetics by various in situ experiments are essential contributions that distinct this study from our prior publication.

Comments 9: I disagree with authors response to my comment 7. Authors are patently wrong in calling their reaction as solution-phase topochemical polymerization. The example cited by the authors in support of their argument (Angew. Chem. Int. Ed.2006, 45, 5383-5386) describes topochemical polymerization of diynes in solid microfibers formed via self-assembly (gelation). That is a solid-state reaction. Also authors own scattering studies and schematic representation also contradict their argument.

Response: While we acknowledge that gelation is a common outcome in the formation of supramolecular framework (e.g., ref 36 and 37), we want to clarify that this is neither the case in our system nor in the reference cited by the Reviewer (ref 33). To fact-check,

we directly quote from reference 33, which described the behavior of the monomers in solution:

“**1** and **2** were prepared by anionic polymerization of isoprene, high-pressure hydrogenation, stepwise solution-phase peptide synthesis, and acetylene heterocoupling reactions.¹⁶ Solutions of **1** and **2** in CH₂Cl₂ or CHCl₃ showed no tendency toward gelation, but ¹H NMR spectra as well as solution-phase IR spectra gave a clear indication of aggregation.¹⁶”

This statement clearly described the formation of supramolecular polymers in solution without gelation, contradicting the Reviewer’s assertion that the reference describes a gel-based solid-state reaction. Furthermore, the solutions of the supramolecular polymers were later photopolymerized, demonstrating the polymerizability of these supramolecular polymers in solution.

Additionally, as previously clarified, our scattering studies pertain to the polymer, not the monomers, and therefore do not contradict our claim.

Comments 10: In response to my comment 8, authors provided Temp-dependent PXRD data. This clearly shows its thermal polymerization reaction at 65 degree C . Even at lower temperature the shoulder peak, presumably due to polymer phase, is seen. This contradicts with authors contention that thermal polymerization does not occur at 333K (60 degree C).

Response: As clarified in our response to Comment 5, we acknowledge that thermal polymerization could occur at lower temperatures. Strictly speaking, according to thermodynamic principles, thermal reactions can take place at any temperature above absolute zero. However, that is meaningless without considering the kinetics. Our experiments have shown that while thermal polymerization at 333 K (60 °C) does occur in the absence of light, it is slow and negligible within 1000 s. This was clarified in our last revision, as stated: “Notably, no appreciable monomer-polymer conversion was observed without light irradiation (Supplementary Figs. S27-29). These results confirm the negligible contribution of thermally induced polymerization at temperatures below 333 K, consistent with the DSC results.” The Reviewer’s interpretation that we claimed “thermal polymerization does not occur at 333 K” may stem from a misunderstanding. Our contention is not that no thermal polymerization occurs, but rather that its contribution is insignificant under the given conditions.

Comments 11: Regarding my comment 9, authors did not explain how X-ray is facilitating the reaction.

Response: Compared to commonly used UV-visible light irradiation, X-ray (and gamma-ray) ionization radiation sources have a higher penetration depth that facilitates more

homogeneous reactions throughout the crystal. More importantly, high-energy photons likely initiate polymerization through secondary processes derived from inelastic scattering or core electron excitations, rather than direct valence electron excitation as seen in UV-visible light irradiation. This makes X-ray irradiation more effective in facilitating polymerization while mitigating crystal disintegration. We have included a related discussion in the revision: “We note that visible light irradiation only led to low quality crystals. Compared to commonly used light irradiation, X-ray (and gamma-ray) ionization radiation sources have a higher penetration depth that facilitates more homogeneous reactions throughout the crystal. More importantly, high-energy photons likely initiate polymerization through secondary processes derived from inelastic scattering or core electron excitations, rather than direct valence electron excitation as seen in UV-visible light irradiation. This makes X-ray irradiation more effective in facilitating polymerization while mitigating crystal disintegration.”

Reviewer #2 (Remarks to the Author):

Comment: The author has addressed my concerns, and this article can now be published.

Response: We thank the Reviewer for the kind considerations.

Reviewer #4 (Remarks to the Author):

Comments: I will express my opinion below regarding reviewer #1's most serious points (second round).

(1) Regarding the phrase “topochemical polymerization in solution” (General Comment and comment 7):

I agree with reviewer #1’s opinion that “topochemical polymerization in solution” will cause a serious misleading. As described in this manuscript and indicated by reviewer #1, topochemical reaction including polymerization proceeds in the solid state. Obviously, topochemical polymerization of monomers molecularly dissolved in solution is impossible. The polymerizations described here are the results of the polymerization in the solid-state of supramolecular aggregates of AQM monomers in a specific solvent mixture (THF and water) as schematically shown in Fig. 5a, and the mixture happened to be transparent, so the authors thought it was a homogeneous solution (it is not). Most importantly, topochemical polymerization occurs in the solid, not in solution. To avoid misleading, the authors should revise “the solution phase”, “solution-phase polymerization”, “solution-based TCP”, and etc. in the manuscript. For example, “topochemical polymerization in

the solid-state of supramolecular aggregates dispersed in (transparent) solution” seems to be appropriate.

Response: We thank the Reviewer for the kind suggestions. We have made changes accordingly to indicate that these reactions take place in a liquid medium, to avoid potential confusion.

Comments: It seems worthwhile to perform experiments to measure the light scattering of "supramolecular aggregates dispersed in a (transparent) solution" and to estimate the size of the aggregates.

Response: We thank the Reviewer for the valuable insight. Following the suggestion, we conducted dynamic light scattering (DLS) studies to estimate the aggregate sizes of CM1-s and CM2-s. As shown in Figure R1, the average hydrodynamic diameters (D_h) of CM1-s and CM2-s were determined to be ~ 300 nm and 199 nm, respectively. Following irradiation with 450 nm light, the D_h of the resulting CP1-s and CP2-s solutions increased to 409 nm and 221 nm, respectively. The increase is consistent with the formation of corresponding polymers (*Poly. Chem.* 2020, 11, 1947). Due to the rapid polymerization kinetics of CM2-s, the broad peak of CM2-s, along with a small side peak centered at $D_h = 33$ nm (1.4%), is attributed to likely spontaneous TCP occurring during the DLS scanning process.

Figure **R1**. Size distribution determined by DLS for CM1-s and CP1-s (a) and CM2-s and CP2-s (b).

We have added the additional discussion in the main text of our revised manuscript (Page 18) and Supplementary Information as Figure S35.

Comments: On the other hand, the most important conclusion of this paper seems to be that it has been discovered that crystalline polymers equivalent to those obtained by

topochemical polymerization of single crystals and thin films can be obtained by preparing supramolecular aggregates by adding a poor solvent (water) to a monomer solution in THF and then polymerizing it. Considering the ripple effects in the future, it is felt that there is value in publishing it in this journal after revision according to the above comments.

Response: We thank the Reviewer for the kind considerations.

Comments: (2) Regarding the topochemical polymerization mechanism (comments 8 and 9):

To dispel the doubts and concerns that reviewer #1 has about the topochemical polymerization mechanism, the authors should provide more convincing evidence that it is not thermal polymerization.

Response: We thank the Reviewer for the suggestion. We would like to note that Reviewer 1's interpretation of our contention as "thermal polymerization does not occur at 333 K" is likely based on misunderstanding. We'd like to clarify that it is not meaningful to rule out spontaneous thermal reaction. As thermodynamic principles dictate, some level of thermal reaction is inevitable at non-zero temperatures. However, it would only become relevant if the thermal dark reaction is fast enough to contribute significantly to the overall reaction rate. We acknowledge that, as shown by our experiments, thermal polymerization at 333 K in the absence of light may occur but is at a much slower rate, thus is negligible, as supported by Figures S27-S29 where there was no change of absorption within ~1000 s at 333 K. At room temperature, no change of PXRD spectra was observed over a period of 8 hrs (Figure S12a). This had been clarified in our last revision, as quoted below: "Notably, no appreciable monomer-polymer conversion was observed without light irradiation (Supplementary Figs. S27-29). These results confirm the negligible contribution of thermally induced polymerization at temperatures below 333 K, consistent with the DSC results."

Point-to-point responses

Reviewer #4 (Remarks to the Author):

The authors performed additional measurements such as DLS and revised the manuscript and Supporting Information based on the new results, which certainly improved the quality of this manuscript. Most of my concerns were also addressed by the authors. However, the misleading phrases, such as “solution-based TCP” (line 77), “aqueous solutions” (line 87), “TCP of AQMs in solution” (line 385), “in liquid” (line 386), “The resulting solutions” (lines 467-468), and “solution-mediated TCP” (lines 469-470) etc in the revised manuscript have not been sufficiently corrected. There may be other misleading phrases in the revised manuscript and Supporting Information. The authors should carefully correct and/or rephrase as suggested before such as “Most importantly, topochemical polymerization occurs in the solid, not in solution. To avoid misleading, the authors should revise “the solution phase”, “solution-phase polymerization”, “solution-based TCP”, and etc. in the manuscript. For example, “topochemical polymerization in the solid-state of supramolecular aggregates dispersed in (transparent) solution” seems to be appropriate.”

This reviewer may then recommend publication.

Response: We fully appreciate the constructive and suggestive comments from Reviewer#4. In accordance with the suggestions, we have appropriately addressed the raised concerns and highlighted them in red in the manuscript for clarity:

1. Changing “solution-based TCP” (line 77) into “TCP in liquid-medium”;
2. Changing “aqueous solutions” (line 87) into “aggregates in solution”;
3. Changing “TCP of AQMs in solution” (line 385) into “TCP of AQMs in aggregates”;
4. Changing “in liquid” (line 386) into “in aggregates”;
5. Changing “The resulting solutions” (lines 467-468) into “The resulting aggregates dispersed in transparent solution”;
6. Changing “solution-mediated TCP” (lines 469-470) into “TCP process”.
7. As suggested, we have added the sentence “Notably, TCP occurs within the aggregates dispersed in the transparent solution rather than in the fully dissolved monomeric state.” in the manuscript.
8. We also carefully checked SI and changed corresponding description in the legends of Fig. S33 and S34.